# Nonasymptotic Analysis of Stochastic Gradient Descent with the Richardson–Romberg Extrapolation

**Marina Sheshukova**[1,5]  **Denis Belomestny**[2,1]  **Alain Durmus** [3]  **Eric Moulines**[3,4]
**Alexey Naumov**[1,6]  **Sergey Samsonov**[1]
[1]HSE University   [2]Duisburg-Essen University   [3]CMAP, UMR 7641, Ecole Polytechnique
[4]Mohamed Bin Zayed University of AI   [5] Skolkovo Institute of Science and Technology
[6] Steklov Mathematical Institute of Russian Academy of Sciences
{msheshukova,anaumov,svsamsonov}@hse.ru
{alain.durmus,eric.moulines}@polytechnique.edu
denis.belomestny@uni-due.de

## Abstract

We address the problem of solving strongly convex and smooth minimization problems using stochastic gradient descent (SGD) algorithm with a constant step size. Previous works suggested to combine the Polyak-Ruppert averaging procedure with the Richardson-Romberg extrapolation to reduce the asymptotic bias of SGD at the expense of a mild increase of the variance. We significantly extend previous results by providing an expansion of the mean-squared error of the resulting estimator with respect to the number of iterations $n$. We show that the root mean-squared error can be decomposed into the sum of two terms: a leading one of order $\mathcal{O}(n^{-1/2})$ with explicit dependence on a minimax-optimal asymptotic covariance matrix, and a second-order term of order $\mathcal{O}(n^{-3/4})$, where the power $3/4$ is best known. We also extend this result to the higher-order moment bounds. Our analysis relies on the properties of the SGD iterates viewed as a time-homogeneous Markov chain. In particular, we establish that this chain is geometrically ergodic with respect to a suitably defined weighted Wasserstein semimetric.

## 1 Introduction

Stochastic gradient methods are a fundamental approach for solving a wide range of optimization problems, with a broad range of applications including generative modeling (Goodfellow et al., 2014; 2016), empirical risk minimization (Van der Vaart, 2000), and reinforcement learning (Sutton & Barto, 2018; Schulman et al., 2015; Mnih et al., 2015). These methods solve the stochastic minimization problem

$$\min_{\theta \in \mathbb{R}^d} f(\theta), \qquad \nabla f(\theta) = \mathsf{E}_{\xi \sim \mathbb{P}_\xi}[\nabla F(\theta, \xi)], \tag{1}$$

where $\xi$ is a random variable with distribution $\mathbb{P}_\xi$, and the gradient $\nabla f$ of the function $f$ can be accessed only through (unbiased) noisy estimates $\nabla F$. Throughout this paper, we consider strongly convex minimization problems admitting a unique solution $\theta^\star$. Arguably the simplest and one of the most widely used approaches to solve (1) is the stochastic gradient descent (SGD). This algorithm constructs the sequence of updates

$$\theta_{k+1} = \theta_k - \gamma_{k+1} \nabla F(\theta_k, \xi_{k+1}), \quad \theta_0 \in \mathbb{R}^d, \tag{2}$$

where $\{\gamma_k\}_{k \in \mathbb{N}}$ are step sizes, either diminishing or constant, and $\{\xi_k\}_{k \in \mathbb{N}}$ is an i.i.d. sequence with distribution $\mathbb{P}_\xi$. The algorithm (2) can be viewed as a special instance of the Robbins-Monro procedure (Robbins & Monro, 1951). While the SGD algorithm remains one of the core algorithms in statistical inference, its performance can be enhanced by means of additional techniques that use e.g., momentum (Qian, 1999), averaging (Ruppert, 1988; Polyak & Juditsky, 1992), or variance reduction (Defazio et al., 2014; Nguyen et al., 2017). In particular, the celebrated Polyak-Ruppert

algorithm proceeds with a trajectory-wise averaging of the estimates

$$\bar{\theta}_{n_0,n} = \frac{1}{n} \sum_{k=n_0+1}^{n+n_0} \theta_k \tag{3}$$

for some $n_0 > 0$. It is known (Polyak & Juditsky, 1992; Fort, 2015), that under appropriate assumptions on $f$ and $\gamma_k$, the sequence of estimates $\{\bar{\theta}_{n_0,n}\}_{n \in \mathbb{N}}$ is asymptotically normal, that is,

$$\sqrt{n}(\bar{\theta}_{n_0,n} - \theta^\star) \xrightarrow{d} \mathrm{N}(0, \Sigma_\infty), \quad n \to \infty \tag{4}$$

where $\xrightarrow{d}$ denotes the convergence in distribution and $\mathrm{N}(0, \Sigma_\infty)$ denotes the zero-mean Gaussian distribution with covariance matrix $\Sigma_\infty$, which is asymptotically optimal, see Fort (2015) for a discussion. On the other hand, quantitative counterparts of (4) rely on the root mean squared error (root-MSE) bounds of the form

$$\mathsf{E}^{1/2}[\|\bar{\theta}_{n_0,n} - \theta^\star\|^2] \leq \frac{\sqrt{\mathrm{Tr}\,\Sigma_\infty}}{n^{1/2}} + \frac{C(f,d)}{n^{1/2+\delta}} + \mathcal{R}(\|\theta_0 - \theta^\star\|, n). \tag{5}$$

Here $\mathcal{R}(\|\theta_0 - \theta^\star\|, n)$ is a remainder term which reflects the dependence upon initial condition, $C(f,d)$ is some instance-dependent constant and $\delta > 0$. There are many studies establishing (5) for Polyak-Ruppert averaged SGD under various model assumptions, including Moulines & Bach (2011); Gadat & Panloup (2023). In particular, Li et al. (2022) derived the bound (5) with the rate $\delta = 1/4$, which is the best known among first-order methods, but their results apply to a modified two-timescale algorithm with multiple restarts (Root-SGD). In our work, we show that the same upper bound is achieved by a simple modification of the estimate $\bar{\theta}_{n_0,n}$ based on Richardson-Romberg extrapolation. The main contributions of the current paper are as follows:

- We show that a version of SGD algorithm with constant step size, Polyak-Ruppert averaging, and Richardson-Romberg extrapolation lead to the root-MSE bound (5) with $\delta = 1/4$ when applied to strongly convex minimization problems. We obtain this result by leveraging the analysis of iterates of the constant step-size SGD as a Markov chain. It is important to note that this result is obtained for a fixed step size $\gamma$ of order $1/\sqrt{n}$ with $n$ being a total number of iterations. This result requires that the number of samples $n$ is known a priori to optimize the step size $\gamma$.

- We generalize the above result and obtain high-order moment bounds on the error. Selecting the constant step size $\gamma = 1/\sqrt{n}$, we obtain for $p \geq 2$ bounds of the form

$$\mathsf{E}^{1/p}[\|\bar{\theta}_n^{(RR)} - \theta^\star\|^p] \leq \frac{c_0 p^{1/2}\sqrt{\mathrm{Tr}\,\Sigma_\infty}}{n^{1/2}} + \frac{C(f,d,p)}{n^{3/4}} + \mathcal{R}(\|\theta_0 - \theta^\star\|, n, p), \tag{6}$$

where $c_0$ is a universal constant, and $\bar{\theta}_n^{(RR)}$ is a counterpart of $\bar{\theta}_{n_0,n}$ when using Richardson-Romberg extrapolation, see related definitions in Section 4. Our proof is based on a novel version of the Rosenthal inequality, which might be of independent interest.

The rest of the paper is organized as follows. We provide a literature review on the non-asymptotic analysis of the SGD algorithm and its modifications, with an emphasis on constant step-size SGD and the Richardson-Romberg procedure in Section 2. Next, we analyze constant step-size SGD by treating it as a Markov chain and study the properties of the Polyak-Ruppert averaged estimator (3) in Section 3. In Section 4, we analyze the properties of Richardson-Romberg extrapolation applied to Polyak-Ruppert averaged SGD and derive bounds for the second-order and higher-order moments of the error.

**Notations and definitions.** For $\theta_1, \ldots, \theta_k$ being the iterates of stochastic first-order method, we denote $\mathcal{F}_k = \sigma(\theta_0, \theta_1, \ldots, \theta_k)$. Let $(\mathsf{Z}, \mathsf{d}_\mathsf{Z})$ be a complete separable metric space equipped with its Borel $\sigma$-algebra $\mathcal{Z}$. We call a function $c : \mathsf{Z} \times \mathsf{Z} \to \mathbb{R}_+$ a *distance-like* function, if it is symmetric, lower semi-continuous and $c(x,y) = 0$ if and only if $x = y$, and there exists $q \in \mathbb{N}$ such that $\mathsf{d}_\mathsf{Z}^q(x,y) \leq c(x,y)$. We denote by $\mathscr{C}(\xi, \xi')$ the set of couplings of probability measures $\xi$ and $\xi'$, that is, a set of probability measures on $(\mathsf{Z}^2, \mathcal{Z}^{\otimes 2})$, such that for any $\Pi \in \mathscr{C}(\xi, \xi')$ and any $A \in \mathcal{Z}$ it holds $\Pi(\mathsf{Z} \times A) = \xi'(A)$ and $\Pi(A \times \mathsf{Z}) = \xi(A)$. We define the Wasserstein semimetric associated to the distance-like function $c(\cdot, \cdot)$, as

$$\mathbf{W}_c(\xi, \xi') = \inf_{\Pi \in \mathscr{C}(\xi, \xi')} \int_{\mathsf{Z} \times \mathsf{Z}} c(z, z') \Pi(\mathrm{d}z, \mathrm{d}z'). \tag{7}$$

Note that $\mathbf{W}_c(\xi, \xi')$ is not necessarily a distance, as it may fail to satisfy the triangle inequality. In the particular case of $\mathsf{Z} = \mathbb{R}^d$, and $c_p(x, y) = \|x - y\|^p$, $x, y \in \mathbb{R}^d$, $p \geq 1$, we use a short notation for $\mathbf{W}_p(\xi, \xi')$ defined by $\mathbf{W}_p^p(\xi, \xi') = \mathbf{W}_{c_p}(\xi, \xi')$. For $x, y \in \mathbb{R}^d$ denote by $x \otimes y$ the tensor product of $x$ and $y$ and by $x^{\otimes k}$ the $k$-th tensor power of $x$. In addition, for a function $f : \mathbb{R}^d \to \mathbb{R}$ we denote by $\nabla^k f(\theta)$ the $k$-th differential of $f$, that is $\nabla^k f(\theta)_{i_1, \cdots, i_k} = \frac{\partial^k f}{\prod_{j=1}^k \partial \theta_{i_j}}$. For any tensor $M \in (\mathbb{R}^d)^{\otimes (k-1)}$, we define $\nabla^k f(\theta) M \in \mathbb{R}^d$ by the relation $(\nabla^k f(\theta) M)_l = \sum_{i_1, \ldots, i_{k-1}} M_{i_1, \cdots, i_{k-1}} \nabla^k f(\theta)_{i_1, \cdots, i_{k-1}, l}$, where $l \in \{1, \ldots, d\}$. Also for any tensor $M \in (\mathbb{R}^d)^{\otimes (k-1)}$ we define $\|M\| = \sup\limits_{x^l \neq 0, l \in \{1, \ldots, k\}} \frac{\sum_{i_1, \ldots, i_k} M_{i_1, \ldots, i_k} x_{i_1}^1 \cdot \ldots \cdot x_{i_k}^k}{\|x^1\| \cdot \ldots \cdot \|x^k\|}$. For two sequences $\{a_n\}_{n \in \mathbb{N}}$ and $\{b_n\}_{n \in \mathbb{N}}$ we write $a_n \lesssim b_n$, if there is an absolute constant $c > 0$, such that $a_n \leq c b_n$ for any $n \in \mathbb{N}$. Throughout this paper we use $c_0$ for an absolute constant, which values may vary from line to line.

## 2 LITERATURE REVIEW

Constant step-size SGD has been widely studied in the literature. Its bias and MSE were studied for strongly convex problems in (Dieuleveut et al., 2020), for both the last iterate and the Polyak-Ruppert averaging. Yu et al. (2021) studied the bias and the asymptotic normality of the last iterate of SGD for non-convex problems under the Polyak-Lojasiewicz condition and its generalizations. Vlatakis-Gkaragkounis et al. (2024) considered constant step-size methods for solving variational inequalities, focusing on characterizing the bias and establishing asymptotic normality. Merad & Gaïffas (2023) studied the convergence of constant step-size SGD iterates in $\mathbf{W}_p$ distance, $p \geq 1$, and provided concentration bounds for both $\theta_n$ and $\bar{\theta}_{n_0, n}$. Moulines & Bach (2011) derived the root-MSE bound (5) with $\delta = 1/6$ for the case of strongly convex functions $f$. Gadat & Panloup (2023) derived an MSE counterpart to (5) of the form

$$\mathsf{E}[\|\bar{\theta}_{n_0, n} - \theta^\star\|^2] \leq \frac{\mathrm{Tr}\,\Sigma_\infty}{n} + \frac{C(f, d)}{n^{1+\beta}}$$

with $\beta = 1/4$, translating to the root-MSE bound (5) with $\delta = 1/8$. Li et al. (2022) suggested the Root-SGD algorithm combining the ideas of the two-timescale stochastic approximation and multiple restarts, establishing a counterpart of (5) with $\delta = 1/4$. The recent series of papers (Huo et al., 2023; Zhang & Xie, 2024; Zhang et al., 2024) investigated stochastic approximation algorithms with both i.i.d. and Markovian data and constant step sizes. The authors proposed precise characterization of the bias together with an MSE bounds after applying the Richardson-Romberg correction. At the same time, these works only considered the 2-nd moment of the error and did not aim to identify the leading term of the MSE bound, which aligns with the CLT covariance matrix $\Sigma_\infty$.

Richardson-Romberg (RR) extrapolation is a technique used to improve the accuracy of numerical approximations (Hildebrand, 1987), such as those from numerical differentiation or integration. It involves using approximations with different step sizes and then extrapolating to reduce the error, typically by removing the leading term in the error expansion. The one-step RR extrapolation was introduced to reduce the discretization error induced by an Euler scheme to simulate stochastic differential equation in Talay & Tubaro (1990), and later generalized for non-smooth functions in Bally & Talay (1996). This technique was extended using multistep discretizations in Pagès (2007). RR extrapolation have been applied to Stochastic Gradient Descent (SGD) methods in Durmus et al. (2016), Merad & Gaïffas (2023) and Huo et al. (2024a), to improve convergence and reduce error in optimization problems, particularly when dealing with noisy or high-variance gradient estimates. Recent papers (Allmeier & Gast, 2024; Zhang & Xie, 2024; Huo et al., 2024a; Kwon et al., 2024) consider applications of RR to different stochastic approximation problems with constant step-size, including $Q$-learning, and single- and two-timescale stochastic approximation.

## 3 FINITE-TIME ANALYSIS OF THE SGD DYNAMICS FOR STRONGLY CONVEX MINIMIZATION PROBLEMS

### 3.1 GEOMETRIC ERGODICITY OF SGD ITERATES

We consider the following assumption on the function $f$ in the minimization problem (1).

**A1.** *The function $f$ is continuously differentiable and $\mu$-strongly convex on $\mathbb{R}^d$, that is, there exists a constant $\mu > 0$, such that for any $\theta, \theta' \in \mathbb{R}^d$, it holds that*

$$\frac{\mu}{2}\|\theta - \theta'\|^2 \leq f(\theta) - f(\theta') - \langle \nabla f(\theta'), \theta - \theta' \rangle. \tag{8}$$

**A2.** *The function $f$ is $4$ times continuously differentiable and $\mathrm{L}_2$-smooth on $\mathbb{R}^d$, that is, there is a constant $\mathrm{L}_2 \geq 0$, such that for any $\theta, \theta' \in \mathbb{R}^d$,*

$$\|\nabla f(\theta) - \nabla f(\theta')\| \leq \mathrm{L}_2 \|\theta - \theta'\|. \tag{9}$$

*Moreover, $f$ has bounded $3$-rd and $4$-th derivatives, that is, there exist $\mathrm{L}_3, \mathrm{L}_4 \geq 0$ such that*

$$\|\nabla^i f(\theta)\| \leq \mathrm{L}_i \text{ for } i \in \{3, 4\}. \tag{10}$$

We aim to solve the problem (1) using SGD with a constant step size, starting from initial distribution $\nu$. That is, for $k \geq 0$ and a step size $\gamma \geq 0$, we consider the following recurrent scheme

$$\theta_{k+1}^{(\gamma)} = \theta_k^{(\gamma)} - \gamma \nabla F(\theta_k^{(\gamma)}, \xi_{k+1}), \quad \theta_0^{(\gamma)} = \theta_0 \sim \nu, \tag{11}$$

where $\{\xi_k\}_{k \in \mathbb{N}}$ is a sequence satisfying the following condition.

**A3** (p). *$\{\xi_k\}_{k \in \mathbb{N}}$ is a sequence of independent and identically distributed (i.i.d.) random variables with distribution $\mathbb{P}_\xi$, such that $\xi_i$ and $\theta_0$ are independent and for any $\theta \in \mathbb{R}^d$ it holds that*

$$\mathsf{E}_{\xi \sim \mathbb{P}_\xi}[\nabla F(\theta, \xi)] = \nabla f(\theta).$$

*Moreover, there exists $\tau_p$, such that $\mathsf{E}^{1/p}[\|\nabla F(\theta^\star, \xi)\|^p] \leq \tau_p$, and for any $q = 2, \ldots, p$ it holds with some $\mathrm{L}_1 > 0$ that for any $\theta_1, \theta_2 \in \mathbb{R}^d$,*

$$\mathrm{L}_1^{q-1}\|\theta_1 - \theta_2\|^{q-2}\langle \nabla f(\theta_1) - \nabla f(\theta_2), \theta_1 - \theta_2 \rangle \geq \mathsf{E}_{\xi \sim \mathbb{P}_\xi}[\|\nabla F(\theta_1, \xi) - \nabla F(\theta_2, \xi)\|^q]. \tag{12}$$

Assumption **A3**($p$) generalizes the well-known $\mathrm{L}_1$-co-coercivity assumption, see Dieuleveut et al. (2020). A sufficient condition which allows for **A3**($p$) is to assume that $F(\theta, \xi)$ is $\mathbb{P}_\xi$-a.s. convex with respect to $\theta \in \mathbb{R}^d$. For ease of notation, we set

$$\mathrm{L} = \max(\mathrm{L}_1, \mathrm{L}_2, \mathrm{L}_3, \mathrm{L}_4), \tag{13}$$

and trace only dependence upon $\mathrm{L}$ in our subsequent bounds. In this paper we focus on the convergence to $\theta^\star$ of the Polyak-Ruppert averaging estimator defined for any $n \geq 0$,

$$\bar{\theta}_n^{(\gamma)} = \frac{1}{n} \sum_{k=n+1}^{2n} \theta_k^{(\gamma)}. \tag{14}$$

Many previous studies instead consider $\bar{\vartheta}_n^{(\gamma)} = \frac{1}{n-n_0} \sum_{k=n_0+1}^{n} \theta_k^{(\gamma)}$ rather than $\bar{\theta}_n^{(\gamma)}$, where $n \geq n_0 + 1$ and $n_0$ denotes a burn-in period. However, when the sample size $n$ is sufficiently large, the choice of the optimal burn-in size $n_0$ affects the leading terms in the MSE bound of $\bar{\theta}_n^{(\gamma)} - \theta^\star$ only by a constant factor. Therefore, we focus on (14), or equivalently, use $2n$ observations and set $n_0 = n$.

**Properties of $\{\theta_k^{(\gamma)}\}_{k \in \mathbb{N}}$ viewed as a Markov chain.** Under assumptions **A1**, **A2** and **A3**(2), the sequence $\{\theta_k^{(\gamma)}\}_{k \in \mathbb{N}}$ defined by the relation (11) is a time-homogeneous Markov chain with the Markov kernel

$$\mathsf{Q}_\gamma(\theta, \mathsf{A}) = \int_{\mathbb{R}^d} \mathbb{1}_\mathsf{A}(\theta - \gamma \nabla F(\theta, z))\mathsf{P}_\xi(\mathrm{d}z), \quad \theta \in \mathbb{R}^d, \mathsf{A} \in \mathsf{B}(\mathbb{R}^d), \tag{15}$$

where $\mathsf{B}(\mathbb{R}^d)$ is a Borel $\sigma$-field of $\mathbb{R}^d$. In Dieuleveut et al. (2020) it has been established that, under the stated assumptions, $\mathsf{Q}_\gamma$ admits a unique invariant distribution $\pi_\gamma$, if $\gamma$ is small enough. Previous studies, such as Dieuleveut et al. (2020) or Merad & Gaïffas (2023), studied the convergence of the distributions of $\{\theta_k^{(\gamma)}\}_{k \in \mathbb{N}}$ to $\pi_\gamma$ in the $p$-Wasserstein distance $\mathbf{W}_p$, $p \geq 1$, associated with the Euclidean distance in $\mathbb{R}^d$. Our main results require to switch to the non-standard distance-like function, which is defined under **A1** and **A3**(2) as follows:

$$c(\theta, \theta') = \|\theta - \theta'\| \left( \|\theta - \theta^*\| + \|\theta' - \theta^*\| + \frac{2\sqrt{2}\tau_2\sqrt{\gamma}}{\sqrt{\mu}} \right), \quad \theta, \theta^\star \in \mathbb{R}^d. \tag{16}$$

Here the constants $\tau_2$ and $\mu$ are given in **A3**(2) and **A1**, respectively. This distance-like function is designed to analyze $\{\theta_k^{(\gamma)}\}_{k \in \mathbb{N}}$ under **A1** and **A3**(2). In particular, it depends on the step size $\gamma$ and $\theta^\star$. Our first main result establishes *geometric ergodicity* of the Markov kernel $\mathsf{Q}_\gamma$ with respect to the distance-like function $c$ from (16).

**Proposition 1.** *Assume A1, A2, and A3(2). Then for any $\gamma \in (0; 1/(2\,\mathsf{L})]$, the Markov kernel $\mathsf{Q}_\gamma$ defined in (15) admits a unique invariant distribution $\pi_\gamma$. Moreover, $\mathsf{Q}_\gamma$ is geometrically ergodic with respect to the cost function $c$, that is, for any initial distribution $\nu$ on $\mathbb{R}^d$ and $k \in \mathbb{N}$,*

$$\mathbf{W}_c(\nu \mathsf{Q}_\gamma^k, \pi_\gamma) \leq 4(1/2)^{k/m(\gamma)} \mathbf{W}_c(\nu, \pi_\gamma), \tag{17}$$

*where $m(\gamma) = \lceil 2 \log 4/(\gamma \mu) \rceil$.*

**Discussion.** The proof of Proposition 1 is provided in Appendix A.1. Properties of the invariant distribution $\pi_\gamma$ were previously studied in literature, see e.g. Dieuleveut et al. (2020). It particular, it is known (Dieuleveut et al., 2020, Lemma 13), that the 2-nd moment of $\pi_\gamma$ scales linearly with $\gamma$:

$$\int_{\mathbb{R}^d} \|\theta - \theta^\star\|^2 \pi_\gamma(\mathrm{d}\theta) \lesssim \frac{\gamma \tau_2}{\mu}. \tag{18}$$

This bound yields, using Lyapunov's inequality, that

$$\int_{\mathbb{R}^d \times \mathbb{R}^d} \|\theta - \theta'\| \pi_\gamma(\mathrm{d}\theta) \pi_\gamma(\mathrm{d}\theta') \lesssim \sqrt{\frac{\gamma \tau_2}{\mu}}.$$

At the same time, expectation of the cost function $c(\theta, \theta')$ scales linearly with the step size $\gamma$:

$$\int_{\mathbb{R}^d} c(\theta, \theta') \pi_\gamma(\mathrm{d}\theta) \pi_\gamma(\mathrm{d}\theta') \lesssim \frac{\gamma \tau_2}{\mu}. \tag{19}$$

The property (19) is crucial to obtain tighter (with respect to the step size $\gamma$) error bounds for the Richardson-Romberg estimator, as well as in the Rosenthal inequality for additive functional of $\{\theta_k^{(\gamma)}\}_{k \in \mathbb{N}}$ derived in Proposition 8. Precisely, the additional $\sqrt{\gamma}$ factor obtained in (19) would allow us to obtain sharper bounds on the remainder terms in Theorem 6. Next, we analyze the error $\theta_\infty^{(\gamma)} - \theta^\star$ where $\theta_\infty^{(\gamma)}$ is distributed according to the stationary distribution $\pi_\gamma$. To this end, we consider the following condition.

**C1** (p). *There exist constants $\mathsf{D}_{\mathrm{last},p}, \mathsf{C}_{\mathrm{step},p} \geq 2$ depending only on $p$, such that for any step size $\gamma \in (0, 1/(\mathsf{L}\,\mathsf{C}_{\mathrm{step},p})]$, and any initial distribution $\nu$ it holds that*

$$\mathsf{E}_\nu^{2/p}\big[\|\theta_k^{(\gamma)} - \theta^\star\|^p\big] \leq (1 - \gamma \mu)^k \mathsf{E}_\nu^{2/p}\big[\|\theta_0 - \theta^\star\|^p\big] + \mathsf{D}_{\mathrm{last},p} \gamma \tau_p^2/\mu. \tag{20}$$

*Moreover, for the stationary distribution $\pi_\gamma$ it holds that*

$$\mathsf{E}_{\pi_\gamma}^{2/p}\big[\|\theta_\infty^{(\gamma)} - \theta^\star\|^p\big] \leq \mathsf{D}_{\mathrm{last},p} \gamma \tau_p^2/\mu. \tag{21}$$

It is important to note that **C**1 is not independent from the preceding assumptions **A**1 - **A**3(p). In particular, Dieuleveut et al. (2020, Lemma 13) establishes that, under **A**1,**A**2, and **A**3(p), the bound (21) holds for $\gamma \in (0, 1/(\mathsf{L}\,\mathsf{C}_{\mathrm{step},p})]$ with some constants $\mathsf{D}_{\mathrm{last},p}$ and $\mathsf{C}_{\mathrm{step},p}$ depending only upon $p$. Unfortunately, it is difficult to obtain precise dependence of $\mathsf{C}_{\mathrm{step},p}$ and $\mathsf{D}_{\mathrm{last},p}$ on $p$, and to obtain (21) with precise numerical constants. Existing studies (Gadat & Panloup, 2023; Merad & Gaïffas, 2023) either use different set of assumptions or do not explicitly characterize their dependence on $p$. That is why we prefer to state **C**1(p) as a separate assumption. In the subsequent bounds we use **C**1(p) together with **A**1,**A**2, and **A**3(p), tracking the dependence of our bounds upon $\mathsf{C}_{\mathrm{step},p}$ and $\mathsf{D}_{\mathrm{last},p}$. We leave the derivation of **C**1(p) with precise constants $\mathsf{D}_{\mathrm{last},p}, \mathsf{C}_{\mathrm{step},p}$ as an interesting direction for future research.

Under assumption **C**1, we control the fluctuations of $\theta_k^{(\gamma)}$ around $\theta^\star$. However, unless $f$ is quadratic, it is known that $\int_{\mathbb{R}^d} \theta \pi_\gamma(\mathrm{d}\theta) \neq \theta^\star$. In the next proposition, we quantify this bias under under weaker assumptions than those in Dieuleveut et al. (2020, Theorem 4).

**Proposition 2.** *Assume A1, A2, A3(6), and C1(6). Then there exist such $\Delta_1 \in \mathbb{R}^d, \Delta_2 \in \mathbb{R}^{d \times d}$, not depending upon $\gamma$, that for any $\gamma \in (0, 1/(\mathsf{L}\,\mathsf{C}_{\mathrm{step},6})]$, it holds*

$$\bar{\theta}_\gamma := \int_{\mathbb{R}^d} \theta \pi_\gamma(\mathrm{d}\theta) = \theta^\star + \gamma \Delta_1 + B_1 \gamma^{3/2}, \tag{22}$$

$$\bar{\Sigma}_\gamma := \int_{\mathbb{R}^d} (\theta - \theta^\star)^{\otimes 2} \pi_\gamma(\mathrm{d}\theta) = \gamma \Delta_2 + B_2 \gamma^{3/2}. \tag{23}$$

Here $B_1 \in \mathbb{R}^d$ and $B_2 \in \mathbb{R}^{d \times d}$ satisfy $\|B_1\| \leq \frac{L}{\mu}\mathsf{C}_1 + \frac{L\mathsf{D}_{\mathrm{last},6}^{3/2}\tau_6^3}{2\mu^{5/2}}$, $\|B_2\| \leq \mathsf{C}_1$, where $\mathsf{C}_1$ defined in (52) is a constant independent of $\gamma$. Moreover, for any initial distribution $\nu$ on $\mathbb{R}^d$, it holds that

$$\mathsf{E}_\nu[\bar{\theta}_n^{(\gamma)}] = \theta^\star + \gamma\Delta_1 + B_1\gamma^{3/2} + \mathcal{R}_1(\theta_0 - \theta^\star, \gamma, n)\,, \tag{24}$$

where

$$\|\mathcal{R}_1(\theta_0 - \theta^\star, \gamma, n)\| \lesssim \frac{e^{-\gamma\mu(n+1)/2}}{n\gamma\mu}\left(\mathsf{E}_\nu^{1/2}\big[\|\theta_0 - \theta^\star\|^2\big] + \frac{\sqrt{\gamma}\tau_2}{\sqrt{\mu}}\right)\,. \tag{25}$$

The proof is provided in Appendix A. Results of this type are known in the literature for stochastic approximation algorithms, see e.g. Huo et al. (2024b) and Allmeier & Gast (2024). The additive term $\Delta_1$ vanishes when the function $f$ is quadratic, see (Moulines & Bach, 2011).

## 3.2 Analysis of the Polyak-Ruppert averaged estimator $\bar{\theta}_n^{(\gamma)}$.

In this section, we analyze the finite-sample properties of the estimator $\bar{\theta}_n^{(\gamma)}$ from (14). The analysis is based on techniques previously used in Moulines & Bach (2011), as well as in the analysis of the Polyak-Ruppert averaged LSA (Linear Stochastic Approximation) algorithms, see Mou et al. (2020); Durmus et al. (2024). Below we derive the key representation for the error $\bar{\theta}_n^{(\gamma)} - \theta^\star$, following Moulines & Bach (2011). Define the $k$-th step noise level at the point $\theta \in \mathbb{R}^d$ by:

$$\varepsilon_k(\theta) = \nabla F(\theta, \xi_k) - \nabla f(\theta)\,, \tag{26}$$

and $\varepsilon_{k+1}(\theta_k^{(\gamma)})$ is a martingale-difference sequence w.r.t. $(\mathcal{F}_k)_{k \in \mathbb{N}}$. Then the recurrence (11) takes form

$$\theta_{k+1}^{(\gamma)} - \theta^\star = \theta_k^{(\gamma)} - \theta^\star - \gamma\big(\nabla f(\theta_k^{(\gamma)}) + \varepsilon_{k+1}(\theta_k^{(\gamma)})\big)\,. \tag{27}$$

We set

$$\eta(\theta) = \nabla f(\theta) - \mathrm{H}^\star(\theta - \theta^\star)\,, \quad \text{where } \mathrm{H}^\star = \nabla^2 f(\theta^\star) \in \mathbb{R}^{d \times d}\,. \tag{28}$$

We obtain from (27) with simple algebra that

$$\mathrm{H}^\star(\theta_k^{(\gamma)} - \theta^\star) = \gamma^{-1}(\theta_k^{(\gamma)} - \theta_{k+1}^{(\gamma)}) - \varepsilon_{k+1}(\theta_k^{(\gamma)}) - \eta(\theta_k^{(\gamma)})\,. \tag{29}$$

Taking average of (29) for $k = n+1$ to $2n$, we arrive at the final representation:

$$\mathrm{H}^\star(\bar{\theta}_n^{(\gamma)} - \theta^\star) = \frac{\theta_{n+1}^{(\gamma)} - \theta^\star}{\gamma n} - \frac{\theta_{2n+1}^{(\gamma)} - \theta^\star}{\gamma n} - \frac{1}{n}\sum_{k=n+1}^{2n}\varepsilon_{k+1}(\theta_k^{(\gamma)}) - \frac{1}{n}\sum_{k=n+1}^{2n}\eta(\theta_k^{(\gamma)})\,. \tag{30}$$

We further introduce the covariance matrix of $\varepsilon_k(\theta^\star)$ measured at the optimal point $\theta^\star$, that is,

$$\Sigma_\varepsilon^\star = \mathsf{E}_{\xi \sim \mathbb{P}_\xi}[\nabla F(\theta^\star, \xi)^{\otimes 2}]\,. \tag{31}$$

Note that $\Sigma_\varepsilon^\star$ does not depend on the step size $\gamma$ and is related to the asymptotically optimal covariance matrix of the Polyak-Ruppert averaged iterates $\bar{\theta}_n^{(\gamma)}$, see Fort (2015). Precisely, under assumptions A1-A3, the asymptotic covariance matrix $\Sigma_\infty$ from (4) is given by

$$\Sigma_\infty = \{\mathrm{H}^\star\}^{-1}\Sigma_\varepsilon^\star\{\mathrm{H}^\star\}^{-1}\,. \tag{32}$$

We state our subsequent results in terms of $\mathrm{H}^\star(\bar{\theta}_n^{(\gamma)} - \theta^\star)$ and its Richardson-Romberg adjusted counterpart. One can switch to the corresponding results for $\bar{\theta}_n^{(\gamma)} - \theta^\star$ at the price of the factor $\|\{\mathrm{H}^\star\}^{-1}\|$, which affects the non-leading terms. In our first result below, we establish the root MSE bound on the error of the Polyak-Ruppert averaged estimator (14).

**Theorem 3.** *Assume A1, A2, A3(6), and C1(6). Then for any $\gamma \in (0, 1/(\mathrm{L}\,\mathsf{C}_{\mathrm{step},6})]$, $n \in \mathbb{N}$, and initial distribution $\nu$ on $\mathbb{R}^d$, the sequence of Polyak-Ruppert estimates (14) satisfies*

$$\mathsf{E}_\nu^{1/2}[\|\mathrm{H}^\star(\bar{\theta}_n^{(\gamma)} - \theta^\star)\|^2] \leq \frac{\sqrt{\mathrm{Tr}\,\Sigma_\varepsilon^\star}}{\sqrt{n}} + \frac{\mathsf{C}_2}{\gamma^{1/2}n} + \mathsf{C}_3\gamma + \frac{\mathsf{C}_4\gamma^{1/2}}{n^{1/2}} + \mathcal{R}_2(n, \gamma, \|\theta_0 - \theta^\star\|)\,, \tag{33}$$

*where the constants $\mathsf{C}_2$ to $\mathsf{C}_4$ are defined in Appendix B (see equation (66)), and*

$$\mathcal{R}_2(n, \gamma, \|\theta_0 - \theta^\star\|) = \frac{c_0(1 - \gamma\mu)^{(n+1)/2}\,\mathrm{L}}{\gamma\mu n}\mathsf{E}_\nu^{1/2}\big[\|\theta_0 - \theta^\star\|^2\big]$$

$$+ \frac{c_0\,\mathrm{L}(1 - \gamma\mu)^{n+1}}{2n\gamma\mu}\mathsf{E}_\nu^{1/2}\big[\|\theta_0 - \theta^\star\|^4\big]\,,$$

*where $c_0$ is an absolute (numerical) constant.*

The version of Theorem 3 with explicit constants together with the proof is provided in Appendix B, see Theorem 15. Note that the result of Theorem 3 is valid for arbitrary $\gamma \in (0, 1/(\mathsf{L}\,\mathsf{C}_{\text{step},6})]$. At the same time, this bound can be optimized over step size of the form $\gamma = n^{-\beta}$, $\beta \in (0, 1)$.

**Corollary 4.** *Under the assumptions of Theorem 3, provided that $n \geq (\mathsf{L}\,\mathsf{C}_{\text{step},6})^{3/2}$, it holds setting $\gamma = n^{-2/3}$ that*

$$\mathsf{E}_\nu^{1/2}[\|\,\mathsf{H}^\star(\bar{\theta}_n^{(\gamma)} - \theta^\star)\|^2] \leq \frac{\sqrt{\operatorname{Tr} \Sigma_\varepsilon^\star}}{n^{1/2}} + \frac{\mathsf{C}(\mathsf{L}, \mu)}{n^{2/3}} + \mathcal{R}_2(n, 1/n^{2/3}, \|\theta_0 - \theta^\star\|) \,, \tag{34}$$

*where the expression for $\mathsf{C}(\mathsf{L}, \mu)$ can be traced from Appendix B, eq. (66).*

Corollary 4 implies that, if $n$ is known in advance and $\gamma = n^{-2/3}$, then $\bar{\theta}_n^{(\gamma)}$ satisfies (5) with $\delta = 1/6$. A closer inspection of the sum (30) reveals that $\mathsf{E}_{\pi_\gamma}[\eta(\theta_k^\gamma)]$ is of order $\gamma$, and we can not expect to provide a better bound for the term $\frac{1}{n}\sum_{k=n+1}^{2n} \eta(\theta_k^\gamma)$ compared to the one coming from Minkowski's inequality. Thus, this is the *bias* of the stationary distribution, which does not allows us to improve scaling of the second-order term w.r.t. the sample size $n$.

In case of deterministic problems $\varepsilon_k(\theta) = 0$ for any $k$ and $\theta$, and $\mathbf{C1}(p)$ is satisfied for any $p \geq 2$ with $\mathsf{D}_{\text{last},p} = 0$. In such a setting, $\Sigma_\varepsilon^\star = 0$, and the remainder terms are proportional to $\mathsf{D}_{\text{last},p}$ with $p = 2, 4$, or $6$, and also vanishes. Therefore, Theorem 3 provides exponential convergence bounds, which are embedded in the remainder term. Previous studies in (Moulines & Bach, 2011) provides the bound of the same order $\mathcal{O}(n^{-2/3})$ for the second-order term of the root-MSE bound of SGD algorithm with Polyak-Ruppert averaging. This rate is known to be suboptimal for first-order methods. The recent work by Li et al. (2022) shows that the best known second-order error term in the bound (34) is of order $\mathcal{O}(n^{-3/4})$ and can be achieved by the Root-SGD algorithm. In the next section we mirror this bound using the constant step-size SGD algorithm combined with the Richardson-Romberg extrapolation technique.

## 4 RICHARDSON-ROMBERG EXTRAPOLATION

Our analysis presented in Theorem 3 was based on the summation by parts formula (30) and Taylor expansion of the gradient $\nabla f(\theta)$ in the neighborhood of $\theta^\star$, yielding the remainder quantity $\eta(\theta)$. It is important to notice that

$$\int_{\mathbb{R}^d} \eta(\theta)\pi_\gamma(\mathrm{d}\theta) \neq 0 \,, \tag{35}$$

which prevents us from using larger step size $\gamma$ in the optimized bound (34). In this section we show that Richardson-Romberg extrapolation technique is sufficient to significantly reduce the bias associated with $\eta(\theta)$ and improve the second-order term in the MSE bound (34). Instead of considering a single SGD trajectory $\{\theta_k^{(\gamma)}\}_{k\in\mathbb{N}}$, and then relying on the tail-averaged estimator $\bar{\theta}_n^{(\gamma)}$, we construct two parallel chains based on the same sequence $\{\xi_k\}_{k\in\mathbb{N}}$:

$$\theta_{k+1}^{(\gamma)} = \theta_k^{(\gamma)} - \gamma\nabla F(\theta_k^{(\gamma)}, \xi_{k+1})\,, \quad \bar{\theta}_n^{(\gamma)} = \frac{1}{n}\sum_{k=n+1}^{2n}\theta_k^{(\gamma)}\,,$$

$$\theta_{k+1}^{(2\gamma)} = \theta_k^{(2\gamma)} - 2\gamma\nabla F(\theta_k^{(2\gamma)}, \xi_{k+1})\,, \quad \bar{\theta}_n^{(2\gamma)} = \frac{1}{n}\sum_{k=n+1}^{2n}\theta_k^{(2\gamma)}\,. \tag{36}$$

Based on $\bar{\theta}_n^{(\gamma)}$ and $\bar{\theta}_n^{(2\gamma)}$ defined above, we construct the Richardson-Romberg estimator:

$$\bar{\theta}_n^{(RR)} := 2\bar{\theta}_n^{(\gamma)} - \bar{\theta}_n^{(2\gamma)}\,. \tag{37}$$

Note that it is possible to use different sources of randomness $\{\xi_k\}_{k\in\mathbb{N}}$ and $\{\xi_k'\}_{k\in\mathbb{N}}$ when constructing the sequences $\{\theta_k^{(\gamma)}\}_{k\in\mathbb{N}}$ and $\{\theta_k^{(2\gamma)}\}_{k\in\mathbb{N}}$, respectively. At the same time, it is possible to show the benefits of using the same sequence of random variables $\{\xi_k\}_{k\in\mathbb{N}}$ in (36). Indeed, consider the decomposition (30) and further expand the term $\eta(\theta)$ defined in (28) as

$$\eta(\theta) = \psi(\theta) + G(\theta)\,,$$

where we have defined the following vector-valued functions:

$$\psi(\theta) = \frac{1}{2}\nabla^3 f(\theta^*)(\theta - \theta^\star)^{\otimes 2}\,, \quad G(\theta) = \frac{1}{2}\left(\int_0^1 t^2\nabla^4 f(t\theta^\star + (1-t)\theta)\,\mathrm{d}t\right)(\theta - \theta^\star)^{\otimes 3}\,. \tag{38}$$

We further rewrite the decomposition (30) as

$$
\mathrm{H}^{\star}(\bar{\theta}_n^{(\gamma)} - \theta^{\star}) = \frac{\theta_{n+1}^{(\gamma)} - \theta^{\star}}{\gamma n} - \frac{\theta_{2n+1}^{(\gamma)} - \theta^{\star}}{\gamma n} - \frac{1}{n} \sum_{k=n+1}^{2n} \varepsilon_{k+1}(\theta^{\star})
$$

$$
- \frac{1}{n} \sum_{k=n+1}^{2n} \{\varepsilon_{k+1}(\theta_k^{(\gamma)}) - \varepsilon_{k+1}(\theta^{\star})\} - \frac{1}{n} \sum_{k=n+1}^{2n} \psi(\theta_k^{(\gamma)}) - \frac{1}{n} \sum_{k=n+1}^{2n} G(\theta_k^{(\gamma)}). \quad (39)
$$

In the decomposition (39), the linear term $W = n^{-1} \sum_{k=n+1}^{2n} \varepsilon_{k+1}(\theta^{\star})$ does not depend upon $\gamma$. Moreover, when setting the step size $\gamma = c_0 n^{-\beta}$ with an appropriate $\beta \in (0, 1)$, we can show that the moments of all other terms except for $W$ in the r.h.s. of (39) are small (see Theorem 9 for more details). Hence, using the same sequence $\{\xi_k\}_{k \in \mathbb{N}}$ of noise variables in (36) yields an estimator $\bar{\theta}_n^{(RR)}$, such that its leading component of the variance still equals $W$. Hence, using the Richardson-Romberg procedure increases only the second-order (w.r.t. $n$) components of the variance. At the same time, using different random sequences $\{\xi_k\}_{k \in \mathbb{N}}$ and $\{\xi_k'\}_{k \in \mathbb{N}}$ for $\bar{\theta}_n^{(\gamma)}$ and $\bar{\theta}_n^{(2\gamma)}$ increases the leading component of the MSE by a constant factor. Hence, it is preferable to use synchronous noise construction as in (36). Proposition 2 implies the following improved bound on the bias of $\bar{\theta}_n^{(RR)}$:

**Proposition 5.** *Assume A1, A2, A3(6), and C1(6). Then, for any $\gamma \in (0, 1/(\mathrm{L}\,\mathsf{C}_{\mathrm{step},6})]$, and any initial distribution $\nu$ on $\mathbb{R}^d$, it holds that*

$$
\mathsf{E}_{\nu}[\bar{\theta}_n^{(RR)}] = \theta^{\star} + B_3 \gamma^{3/2} + \mathcal{R}_3(\theta_0 - \theta^{\star}, \gamma, n), \quad (40)
$$

*where $B_3 \in \mathbb{R}^d$ is a vector such that $\|B_3\| \leq \frac{L}{\mu} \mathsf{C}_1 + \frac{L D_{\mathrm{last},6}^{3/2} \tau_6^3}{2\mu^{5/2}}$, and*

$$
\|\mathcal{R}_3(\theta_0 - \theta^{\star}, \gamma, n)\| \lesssim \frac{\mathrm{e}^{-\gamma\mu(n+1)/2}}{n\gamma\mu} \left( \mathsf{E}_{\nu}^{1/2}[\|\theta_0 - \theta^{\star}\|^2] + \frac{\sqrt{\gamma}\tau_2}{\sqrt{\mu}} \right).
$$

The proof of Proposition 5 is provided in Appendix A. This result is a simple consequence of Proposition 5, since the linear in $\gamma$ component of the bias $\gamma\Delta_1$ from (24) cancels out when computing $\bar{\theta}_n^{(RR)}$. We are now ready to formulate the main result for the Richardson-Romberg estimate $\bar{\theta}_n^{(RR)}$.

**Theorem 6.** *Assume A1, A2, A3(6), and C1(6). Then for any $\gamma \in (0, 1/(\mathrm{L}\,\mathsf{C}_{\mathrm{step},6}) \wedge 2/(11\,\mathrm{L})]$, initial distribution $\nu$ and $n \in \mathbb{N}$, the Richardson-Romberg estimator $\bar{\theta}_n^{(RR)}$ defined in (37) satisfies*

$$
\mathsf{E}_{\nu}^{1/2}[\|\mathrm{H}^{\star}(\bar{\theta}_n^{(RR)} - \theta^{\star})\|^2] \leq \frac{\sqrt{\mathrm{Tr}\,\Sigma_{\varepsilon}^{\star}}}{n^{1/2}} + \frac{\mathsf{C}_{\mathrm{RR},1}\gamma^{1/2}}{n^{1/2}} + \frac{\mathsf{C}_{\mathrm{RR},2}}{\gamma^{1/2}n} + \mathsf{C}_{\mathrm{RR},3}\gamma^{3/2} + \frac{\mathsf{C}_{\mathrm{RR},4}\gamma}{n^{1/2}}
$$
$$
+ \mathcal{R}_4(n, \gamma, \|\theta_0 - \theta^{\star}\|),
$$

*where the constants $\mathsf{C}_{\mathrm{RR},1}$ to $\mathsf{C}_{\mathrm{RR},4}$ are defined in Appendix C (equation (74)), and*

$$
\mathcal{R}_4(n, \gamma, \|\theta_0 - \theta^{\star}\|) = \frac{c_0\,\mathrm{L}(1 - \gamma\mu)^{(n+1)/2}}{n\gamma\mu}
$$
$$
\times \left( \mathsf{E}_{\nu}^{1/2}[\|\theta_0 - \theta^{\star}\|^6] + \mathsf{E}_{\nu}^{1/2}[\|\theta_0 - \theta^{\star}\|^4] + \mathsf{E}_{\nu}^{1/2}[\|\theta_0 - \theta^{\star}\|^2] + \frac{\mathsf{D}_{\mathrm{last},4}\gamma\tau_4^2}{\mu} \right),
$$

*with $c_0$ being an absolute constant.*

Proof of Theorem 6 is provided in Appendix C. Similarly to Theorem 3, we can optimize the above bound setting $\gamma$ depending upon $n$.

**Corollary 7.** *Under the assumptions of Theorem 6, provided that $n \geq \mathrm{L}^2(\mathsf{C}_{\mathrm{step},6} \vee 11/2)^2$, it holds setting $\gamma = n^{-1/2}$ that*

$$
\mathsf{E}_{\nu}^{1/2}[\|\mathrm{H}^{\star}(\bar{\theta}_n^{(RR)} - \theta^{\star})\|^2] \leq \frac{\sqrt{\mathrm{Tr}\,\Sigma_{\varepsilon}^{\star}}}{n^{1/2}} + \frac{\mathsf{C}(\mathrm{L}, \mu)}{n^{3/4}} + \mathcal{R}_4(n, 1/\sqrt{n}, \|\theta_0 - \theta^{\star}\|), \quad (41)
$$

*where the expression for $\mathsf{C}(\mathrm{L}, \mu)$ can be traced from Appendix C, eq. (74).*

**Discussion.** Note that the result of Corollary 7 is a counterpart of (5) with $\delta = 1/4$. This decay rate of the second order term is the same as for the Root-SGD algorithm of Li et al. (2022). At the same time, we highlight that the assumptions of Theorem 6 are stronger compared to the ones imposed by Li et al. (2022). In particular, in **A**2 we require that $f$ is 4 times continuously differentiable and uniformly bounded. At the same time, Li et al. (2022) impose Lipschitz continuity of the Hessian of $f$. Our proof of Theorem 6 essentially relies on the 4-th order Taylor expansion, and it is not clear, if this assumption can be relaxed. We leave further investigations of this question for future research.

Now we generalize the previous result for the $p$-th moment bounds with $p \geq 2$. The key technical element of our proof for the $p$-th moment bound is the following statement, which can be viewed as a version of Rosenthal's inequality (Rosenthal, 1970; Pinelis, 1994).

**Proposition 8.** *Let $p \geq 2$ and assume A1, A2, A3(2p), and C1(2p). Then for $\psi$ defined in (38) and any $\gamma \in (0, 1/(\mathrm{L}\,\mathsf{C}_{\mathrm{step},2p})]$, it holds that*

$$\mathsf{E}_{\pi_\gamma}^{1/p}\Big[\|\sum_{k=0}^{n-1}\{\psi(\theta_k^{(\gamma)}) - \pi_\gamma(\psi)\}\|^p\Big] \lesssim \frac{\mathrm{L}\,\mathsf{D}_{\mathrm{last},2p}\,p\tau_{2p}^2\sqrt{n\gamma}}{\mu^{3/2}} + \frac{\mathrm{L}\,\mathsf{D}_{\mathrm{last},2p}\tau_{2p}}{\mu^2}. \tag{42}$$

**Discussion.** Proof of Proposition 8 is provided in Appendix D.1. It is important to acknowledge that there are numerous Rosenthal-type inequalities for dependent sequences in the literature. Proposition 8 can be viewed as an analogue to the classical Rosenthal inequality for strongly mixing sequences, see (Rio, 2017, Theorem 6.3). However, it should be emphasized that the Markov chain $\{\theta_k^{(\gamma)}\}_{k\in\mathbb{N}}$ is geometrically ergodic under the assumptions **A**1-**A**3($p$) only in sense of the weighted Wasserstein semi-metric $\mathbf{W}_c(\xi, \xi')$ with cost function $c$ defined in (16). As a result, the sequence $\{\theta_k^{(\gamma)}\}_{k\in\mathbb{N}}$ does not necessarily satisfy strong mixing conditions. Bounds similar to (42) have been explored in (Durmus et al., 2023), but in Proposition 8 we obtain the bound with tighter dependence of the right-hand side upon $\gamma$. Below we provide the $p$-th moment bound together with corollary for the step size $\gamma$ optimized w.r.t. $n$.

**Theorem 9.** *Let $p \geq 2$ and assume A1, A2, A3(3p), and C1(3p). Then for any step size $\gamma \in (0, 1/(\mathrm{L}\,\mathsf{C}_{\mathrm{step},3p}) \wedge p/(4 \cdot 3^p \,\mathrm{L})]$, initial distribution $\nu$, and $n \in \mathbb{N}$, the estimator $\bar{\theta}_n^{(RR)}$ defined in (37) satisfies*

$$\mathsf{E}_\nu^{1/p}[\|\,\mathrm{H}^\star(\bar{\theta}_n^{(RR)} - \theta^\star)\|^p] \leq \frac{c_1\sqrt{\mathrm{Tr}\,\Sigma_\varepsilon^\star}p^{1/2}}{n^{1/2}} + \frac{\mathsf{C}_{\mathrm{RR},5}}{n\gamma^{1/2}} + \frac{\mathsf{C}_{\mathrm{RR},6}\gamma^{1/2}}{n^{1/2}} + \mathsf{C}_{\mathrm{RR},7}\gamma^{3/2}$$
$$+ \frac{c_2 p\tau_p}{n^{1-1/p}} + \frac{\mathsf{C}_{\mathrm{RR},8}}{n} + \mathcal{R}_5(n, \gamma, \|\theta_0 - \theta^\star\|), \tag{43}$$

*where $c_1 = 60\mathrm{e}$ and $c_2 = 60$ are absolute constants from the Pinelis version of Rosenthal inequality (Pinelis, 1994, Theorem 4.1), and problem-specific constants $\mathsf{C}_{\mathrm{RR},5}$ to $\mathsf{C}_{\mathrm{RR},8}$ are defined in Appendix D (equation (100)), and*

$$\mathcal{R}_5(n, \gamma, \|\theta_0 - \theta^\star\|) = (1-\gamma\mu)^{(n+1)/2}C_{f,p}\big(\mathsf{E}_\nu^{1/p}[\|\theta_0 - \theta^\star\|^p] + \mathsf{E}_\nu^{1/p}[\|\theta_0 - \theta^\star\|^{2p}] + \mathsf{E}_\nu^{1/p}[\|\theta_0 - \theta^\star\|^{3p}]\big).$$

*Here constant $C_{f,p}$ can be traced from Appendix D, eq. (101).*

**Corollary 10.** *Under the assumptions of Theorem 9, provided that $n \geq \mathrm{L}^2\,(\mathsf{C}_{\mathrm{step},3p} \vee 4 \cdot 3^p/p)^2$, it holds setting $\gamma = n^{-1/2}$ that*

$$\mathsf{E}_\nu^{1/p}[\|\,\mathrm{H}^\star(\bar{\theta}_n^{(RR)} - \theta^\star)\|^p] \leq \frac{c_1\sqrt{\mathrm{Tr}\,\Sigma_\varepsilon^\star}p^{1/2}}{n^{1/2}} + \frac{\mathsf{C}(\mathrm{L}, \mu, p)}{n^{3/4}} + \mathcal{R}_5(n, 1/\sqrt{n}, \|\theta_0 - \theta^\star\|), \tag{44}$$

*where the expression for $\mathsf{C}(\mathrm{L}, \mu, p)$ can be traced from Appendix D, eq. (100).*

**Discussion.** Proof of Theorem 9 is provided in Appendix D. Note that the result above is a direct generalization of Theorem 6, which reveals the same scaling of the step size $\gamma$ with respect to $n$. To the best of our knowledge, this is the first analysis of a first-order method, which provides a $p$-th moment bound with $p > 2$ and the second-order term of order $\mathcal{O}(n^{-3/4})$ while keeping the precise leading term related to the asymptotically optimal covariance matrix $\Sigma_\varepsilon^\star$. Such results were previously obtained for the setting of linear stochastic approximation (LSA), see Mou et al. (2020); Durmus et al. (2024). Thus, Richardson-Romberg extrapolation applied to the strongly convex minimization problems allows to mimic the $p$-th moment error bounds that were previously obtained in the LSA setting.

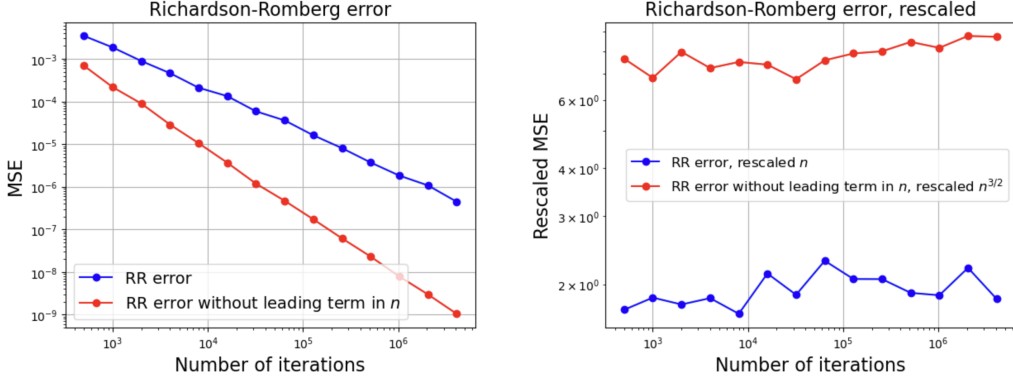

Figure 1: Left picture: Richardson-Romberg experimental error with and without the leading term $\frac{1}{n}\sum_{k=n+1}^{2n}\varepsilon_{k+1}(\theta^{\star})$. Right picture: same errors after rescaling by $n$ and $n^{3/2}$, respectively.

## 5 NUMERICAL RESULTS

In this section we illustrate numerically the scale of the second-order terms in equation (41) in Corollary 7. We show that, for a particular minimization problem, setting $\gamma = n^{-1/2}$, we achieve the scaling of the second-order terms in root-MSE bounds of order $\mathcal{O}(n^{-3/4})$. We consider the problem

$$\min_{\theta\in\mathbb{R}} f(\theta), \quad f(\theta) = \theta^2 + \cos\theta,$$

with the stochastic gradient oracles $\nabla F(\theta, \xi)$ given by $\nabla F(\theta, \xi) = 2\theta - \sin\theta + \xi$, and $\xi \sim \mathcal{N}(0,1)$. This example satisfies the assumptions **A**1, **A**2, **A**3$(p)$ with any $p \geq 2$. We select different sample sizes $n$, choose $\gamma = 1/\sqrt{n}$, and construct the associated estimates $\bar{\theta}_n^{(\gamma)}$ and $\bar{\theta}_n^{(2\gamma)}$. Detailed description of the experimental setting is provided in Appendix E. Then for each $n$ we compute the Richardson-Romberg estimates $\bar{\theta}_n^{(RR)}$ from (36) alongside with its versions without the leading term in $n$, i.e. $\bar{\theta}_n^{(RR)} + n^{-1}\sum_{k=n+1}^{2n}\varepsilon_{k+1}(\theta^{\star})$. We provide first the plot for $\|\bar{\theta}_n^{(RR)} - \theta^{\star}\|^2$ and $\|\bar{\theta}_n^{(RR)} + n^{-1}\sum_{k=n+1}^{2n}\varepsilon_{k+1}(\theta^{\star}) - \theta^{\star}\|^2$, averaged over $M = 320$ parallel runs, in Figure 1. On the same figure we also provide the plots for rescaled errors

$$n\|\bar{\theta}_n^{(RR)} - \theta^{\star}\|^2 \text{ and } n^{3/2}\|\bar{\theta}_n^{(RR)} - \theta^{\star} + n^{-1}\sum_{k=n+1}^{2n}\varepsilon_{k+1}(\theta^{\star})\|^2,$$

also averaged over $M$ parallel runs. The corresponding plot indicates that the proper scaling of the squared norm of the remainder part is $n^{-3/2}$, that is, the corresponding term in root-MSE bound for $\mathsf{E}_\nu^{1/2}[\|\bar{\theta}_n^{(RR)} - \theta^{\star}\|^2]$ scales as $\mathcal{O}(n^{-3/4})$, as predicted by Corollary 7.

## 6 CONCLUSION

In this paper, we study the non-asymptotic error bounds for the Richardson-Romberg estimator built upon the Polyak-Ruppert averaged SGD iterates with a constant step size. In particular, under an appropriate choice of step size, depending on the total number of iterations $n$, the corresponding root-MSE bound admits both a sharp leading term, which aligns with the minimax-optimal covariance matrix, and a second-order term of order $\mathcal{O}(n^{-3/4})$, which is the best known rate among first-order methods. Future research directions include, firstly, generalizing the proposed algorithm to the setting of dependent noise sequences $\{\xi_k\}_{k\in\mathbb{N}}$ in the stochastic gradients (1). Another natural question is to study the properties of $\bar{\theta}_n^{(RR)}$ under relaxed assumptions on $f$. In particular, it would be interesting to remove additional smoothness assumptions on $f$ (bounded 3-rd and 4-th derivatives), and to relax the strong convexity condition **A**1. One more research direction is to quantify a relation between the parameter $\delta$ in (5) and convergence rates in (4), following the approach suggested in Shao & Zhang (2022).

## ACKNOWLEDGEMENT

The work of M. Sheshukova, D. Belomestny, A. Naumov, and S. Samsonov was prepared within the framework of the HSE University Basic Research Program. The work of E. Moulines has been partly funded by the European Union (ERC-2022-SYG-OCEAN-101071601). Views and opinions expressed are however those of the author(s) only and do not necessarily reflect those of the European Union or the European Research Council Executive Agency. Neither the European Union nor the granting authority can be held responsible for them.

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

# A    PROOF OF PROPOSITION 1, PROPOSITION 2, AND PROPOSITION 5

Throughout this appendix we use $c_0$ for an absolute constant, which values may vary from line to line. In addition, when the upper index of $\theta_k^{(\gamma)}$ or $\theta_k^{(2\gamma)}$ is omitted, we assume the result applies to iterations of $\theta_k^{(\gamma)}$. The corresponding results for $\theta_k^{(2\gamma)}$ can be obtained by substituting $2\gamma$ instead of $\gamma$. We provide some additional definitions related to the Markov kernels and kernel couplings, particularly useful when considering convergence in Wasserstein semimetric. Detailed exposition can be found in (Douc et al., 2018, Chapter 20).

Let $Q(z, A)$ be a Markov kernel on $(\mathsf{Z}, \mathcal{Z})$. A Markov kernel K on $(\mathsf{Z}^2, \mathcal{Z}^{\otimes 2})$ is called a kernel coupling of $(Q, Q)$ (that is, of Q with itself), if for all $(z, z') \in \mathsf{Z}^2$ and $A \in \mathcal{Z}$, $\mathrm{K}((z, z'), A \times \mathsf{Z}) = Q(z, A)$ and $\mathrm{K}((z, z'), \mathsf{Z} \times A) = Q(z', A)$. If K is a kernel coupling of $(Q, Q)$, then for all $n \in \mathbb{N}$, $\mathrm{K}^n$ is a kernel coupling of $(Q^n, Q^n)$ and for any $\Pi \in \mathscr{C}(\xi, \xi')$, $\Pi \mathrm{K}^n$ is a coupling of $(\xi Q^n, \xi' Q^n)$. Moreover, it holds that

$$\mathbf{W}_c(\xi Q^n, \xi' Q^n) \leq \int_{\mathsf{Z} \times \mathsf{Z}} \mathrm{K}^n c(z, z') \Pi(\mathrm{d}z \mathrm{d}z'), \tag{45}$$

see (Douc et al., 2018, Corollary 20.1.4). For any probability measure $\Pi$ on $(\mathsf{Z}^2, \mathcal{Z}^{\otimes 2})$, we denote by $\mathbb{P}_\zeta^{\mathrm{K}}$ and $\mathsf{E}_\zeta^{\mathrm{K}}$ the probability and the expectation on the canonical space $((\mathsf{Z}^2)^{\mathbb{N}}, (\mathcal{Z}^{\otimes 2})^{\otimes \mathbb{N}})$ such that the canonical process $\{(Z_n, Z'_n), n \in \mathbb{N}\}$ is a Markov chain with initial probability $\Pi$ and Markov kernel K. We write $\mathsf{E}_{z,z'}^{\mathrm{K}}$ instead of $\mathsf{E}_{\delta_{z,z'}}^{\mathrm{K}}$.

To prove Proposition 1 we need the following auxiliary lemma about the last iterate of SGD algorithm. It can be found in (Dieuleveut et al., 2020, Lemma 10), but we provide its proof here for completeness.

**Lemma 11.** *Assume A1, A2, and A3(2). Then for any $\gamma \in (0; 1/(2\,\mathrm{L})]$ and any $k, r \in \mathbb{N}$ it holds that*

$$\mathsf{E}^{1/2}[\|\theta_{k+r} - \theta^\star\|^2 | \mathcal{F}_k] \leq (1 - \gamma\mu)^{r/2} \|\theta_k - \theta^\star\| + \frac{2^{1/2}\gamma^{1/2}\tau_2}{\mu^{1/2}} \tag{46}$$

*Proof.* Using the recurrence (11), we get

$$\mathsf{E}[\|\theta_{k+1} - \theta^\star\|^2 | \mathcal{F}_k] = \mathsf{E}\big[\|\theta_k - \theta^\star - \gamma\nabla F(\theta_k, \xi_{k+1})\|^2 | \mathcal{F}_k\big]$$
$$= \mathsf{E}\big[\|\theta_k - \theta^\star\|^2 - 2\gamma\langle\nabla F(\theta_k, \xi_{k+1}), \theta_k - \theta^\star\rangle + \gamma^2\|\nabla F(\theta_k, \xi_{k+1})\|^2 | \mathcal{F}_k\big].$$

Applying A3(2), we get

$$\mathsf{E}[\|\theta_{k+1} - \theta^\star\|^2 | \mathcal{F}_k] \leq \|\theta_k - \theta^\star\|^2 - 2\gamma\langle\nabla f(\theta_k) - \nabla f(\theta^\star), \theta_k - \theta^\star\rangle + \gamma^2\mathsf{E}[\|\nabla F(\theta_k, \xi_{k+1})\|^2 | \mathcal{F}_k].$$

Since $\nabla f(\theta^\star) = 0$, using A3(2), we obtain

$$\mathsf{E}[\|\nabla F(\theta_k, \xi_{k+1})\|^2 | \mathcal{F}_k] = \mathsf{E}[\|\nabla F(\theta_k, \xi_{k+1}) - \nabla F(\theta^\star, \xi_{k+1}) + \varepsilon_{k+1}(\theta^\star)\|^2 | \mathcal{F}_k]$$
$$\leq 2\,\mathrm{L}\langle\nabla f(\theta_k) - \nabla f(\theta^\star), \theta_k - \theta^\star\rangle + 2\tau_2^2.$$

Using A1, A2, and the fact that $\gamma \leq 1/(2\,\mathrm{L})$, we get

$$\mathsf{E}[\|\theta_{k+1} - \theta^\star\|^2 | \mathcal{F}_k] \leq (1 - 2\gamma\mu(1 - \mathrm{L}\,\gamma))\|\theta_k - \theta^\star\|^2 + 2\gamma^2\tau_2^2 \leq (1 - \gamma\mu)\|\theta_k - \theta^\star\| + 2\gamma^2\tau_2^2.$$

Hence, applying tower property for conditional expectations, we obtain

$$\mathsf{E}[\|\theta_{k+r} - \theta^\star\|^2 | \mathcal{F}_k] \leq (1 - \gamma\mu)^r \|\theta_k - \theta^\star\|^2 + 2\gamma^2\tau_2^2 \sum_{i=0}^{r}(1 - \gamma\mu)^i \leq (1 - \gamma\mu)^r \|\theta_k - \theta^\star\|^2 + \frac{2\gamma\tau_2^2}{\mu}.$$

$\square$

## A.1    PROOF OF PROPOSITION 1

Consider the synchronous coupling construction defined by the recursions

$$\theta_{k+1}^{(\gamma)} = \theta_k^{(\gamma)} - \gamma\nabla F(\theta_k^{(\gamma)}, \xi_{k+1}), \quad \theta_0^{(\gamma)} = \theta \in \mathbb{R}^d,$$
$$\tilde{\theta}_{k+1}^{(\gamma)} = \tilde{\theta}_k^{(\gamma)} - \gamma\nabla F(\tilde{\theta}_k^{(\gamma)}, \xi_{k+1}), \quad \tilde{\theta}_0^{(\gamma)} = \tilde{\theta} \in \mathbb{R}^d. \tag{47}$$

The pair $(\theta_k^{(\gamma)}, \tilde{\theta}_k^{(\gamma)})_{k \in \mathbb{N}}$ defines a Markov chain with the Markov kernel $\mathrm{K}_\gamma(\cdot, \cdot)$, which is a coupling kernel of $(\mathrm{Q}_\gamma, \mathrm{Q}_\gamma)$. From now on we omit an upper index $(\gamma)$ and write simply $(\theta_k, \tilde{\theta}_k)_{k \in \mathbb{N}}$. Applying now **A**3(2), we get for $\gamma \leq 1/\mathrm{L}$ that

$$\begin{aligned}
\mathsf{E}[\|\theta_{k+1} - \tilde{\theta}_{k+1}\|^2 | \mathcal{F}_k] &= \mathsf{E}[\|\theta_k - \tilde{\theta}_k - \gamma(\nabla F(\theta_k, \xi_{k+1}) - \nabla F(\tilde{\theta}_k, \xi_{k+1}))\|^2 | \mathcal{F}_k] \\
&= \|\theta_k - \tilde{\theta}_k\|^2 + \gamma^2 \mathsf{E}[\|\nabla F(\theta_k, \xi_{k+1}) - \nabla F(\tilde{\theta}_k, \xi_{k+1})\|^2 | \mathcal{F}_k] \\
&\quad - 2\gamma \langle \nabla f(\theta_k) - \nabla f(\tilde{\theta}_k), \theta_k - \tilde{\theta}_k \rangle \\
&\leq (1 - \gamma\mu) \|\theta_k - \tilde{\theta}_k\|^2,
\end{aligned} \tag{48}$$

where in the last inequality we additionally used $1 - 2\gamma\mu(1 - \gamma\mathrm{L}/2) \leq 1 - \gamma\mu$. Similarly, for a cost function $c$ defined in (16), we get using Hölder's and Minkowski's inequalities, that for any $r \in \mathbb{N}$

$$\begin{aligned}
\mathsf{E}[c(\theta_{k+r}, \tilde{\theta}_{k+r}) | \mathcal{F}_k] &\leq \mathsf{E}^{1/2}[\|\theta_{k+r} - \tilde{\theta}_{k+r}\|^2 | \mathcal{F}_k] \big( \mathsf{E}^{1/2}[\|\theta_{k+r} - \theta^\star\|^2 | \mathcal{F}_k] \\
&\quad + \mathsf{E}^{1/2}[\|\tilde{\theta}_{k+r} - \theta^\star\|^2 | \mathcal{F}_k] + \frac{2^{3/2} \gamma^{1/2} \tau_2}{\mu^{1/2}} \big).
\end{aligned}$$

Combining the above inequalities and applying Lemma 11, we obtain

$$\begin{aligned}
\mathsf{E}[c(\theta_{k+r}, \theta'_{k+r}) | \mathcal{F}_k] &\leq (1 - \gamma\mu)^{r/2} \|\theta_k - \tilde{\theta}_k\| \big( (1 - \gamma\mu)^{r/2}(\|\theta_k - \theta^\star\| + \|\tilde{\theta}_k - \theta^\star\|) + \frac{2^{5/2} \gamma^{1/2} \tau_2}{\mu^{1/2}} \big) \\
&\leq 2(1 - \gamma\mu)^{r/2} c(\theta_k, \theta'_k).
\end{aligned}$$

Note that $2(1 - \gamma\mu)^{r/2} \leq 2$ for any $r \leq m(\gamma) - 1$ and $2(1 - \gamma\mu)^{m(\gamma)/2} \leq 1/2$. Hence, applying the result of (Douc et al., 2018, Theorem 20.3.4), we obtain that the Markov kernel $\mathrm{Q}_\gamma$ admits a unique invariant distribution $\pi_\gamma$. Applying (45), we get

$$\mathbf{W}_c(\nu \mathrm{Q}_\gamma^k, \pi_\gamma) \leq 2(1/2)^{\lfloor k/m(\gamma) \rfloor} \mathbf{W}_c(\nu, \pi_\gamma). \tag{49}$$

It remains to note that $(1/2)^{\lfloor k/m(\gamma) \rfloor} \leq 2(1/2)^{k/m(\gamma)}$, and the statement follows.

### A.2 PROOF OF PROPOSITION 2

We first prove (22) and (23) and introduce some additional notations. Under **A**1 – **A**3(2), we define a matrix-valued function $\mathcal{C}(\theta) : \mathbb{R}^d \to \mathbb{R}^{d \times d}$ as

$$\mathcal{C}(\theta) = \mathsf{E}[\varepsilon_1(\theta)^{\otimes 2}]. \tag{50}$$

The result below is essentially based on an appropriate modification of the bounds presented in Dieuleveut et al. (2020, Lemma 18). A careful inspection of its proof reveals that we do not need additional assumptions on $\mathcal{C}(\theta)$, instead we use Lemma 14.

**Lemma 12.** *Assume **A**1, **A**2, **A**3(6), and **C**1(6). Then, for any $\gamma \in (0, 1/(\mathrm{L}\,\mathsf{C}_{\mathrm{step},6})]$, it holds*

$$\bar{\theta}_\gamma - \theta^\star = -(\gamma/2)\{\mathrm{H}^\star\}^{-1}\{\nabla^3 f(\theta^\star)\}\mathbf{T}\mathcal{C}(\theta^\star) + B_1 \gamma^{3/2}, \tag{51}$$

*where $\bar{\theta}_\gamma$ is defined in (22), $\mathcal{C}(\theta)$ is defined in (50), and $B_1 \in \mathbb{R}^d$ satisfies $\|B_1\| \leq \frac{\mathrm{L}}{\mu}\mathsf{C}_1 + \frac{\mathrm{L}\,\mathrm{D}_{\mathrm{last},6}^{3/2}\tau_6^3}{2\mu^{5/2}}$, with*

$$\mathsf{C}_1 = \frac{\sqrt{\mathrm{L}}\tau_2^2}{\sqrt{\mathsf{C}_{\mathrm{step},6}}\mu} + \frac{1}{2}\left( \left( \frac{\mathrm{L}^2 \mathrm{D}_{\mathrm{last},2}}{\mu^{3/2}} + \frac{\mathrm{L}\sqrt{\mathrm{D}_{\mathrm{last},2}}}{\sqrt{\mu}} \right)\tau_2^2 + \frac{\mathrm{L}\,\mathrm{D}_{\mathrm{last},6}^{3/2}\tau_6^3}{2\mu^{3/2}} + \frac{\mathrm{L}^{1/2} \mathrm{D}_{\mathrm{last},4}^2 \tau_4^4}{4\mu^2 \mathsf{C}_{\mathrm{step},6}^{3/2}} \right) \tag{52}$$

*Moreover,*

$$\bar{\Sigma}_\gamma = \gamma \mathbf{T}\mathcal{C}(\theta^\star) + B_2 \gamma^{3/2}, \tag{53}$$

*where the operator $\mathbf{T} : \mathbb{R}^{d \times d} \to \mathbb{R}^{d \times d}$ is defined by the relation*

$$\mathrm{vec}(\mathbf{T}A) = (\mathrm{H}^\star \otimes \mathrm{I} + \mathrm{I} \otimes \mathrm{H}^\star)^{-1} \mathrm{vec}(A)$$

*for any matrix $A \in \mathbb{R}^{d \times d}$, and $B_2 \in \mathbb{R}^{d \times d}$ is a matrix, such that $\|B_2\| \leq \mathsf{C}_1$.*

*Proof.* Let $(\theta_k^{(\gamma)})_{k \in \mathbb{N}}$ be a recurrence defined in (11) with initial distribution $\theta_0 \sim \pi_\gamma$. Recall that $\theta_0$ is independent from the noise variables $(\xi_k)_{k \geq 1}$. First, applying a third-order Taylor expansion of $\nabla f(\theta)$ around $\theta^\star$, we obtain

$$\nabla f(\theta) = \mathrm{H}^\star(\theta - \theta^\star) + (1/2)\{\nabla^3 f(\theta^\star)\}(\theta - \theta^\star)^{\otimes 2} + G(\theta)\,, \tag{54}$$

where $G(\theta)$ has a form

$$G(\theta) = \frac{1}{2}\left(\int_0^1 t^2 \nabla^4 f(t\theta^\star + (1-t)\theta)\, dt\right)(\theta - \theta^\star)^{\otimes 3}\,.$$

Thus, using **A**2,

$$\|G(\theta)\| \leq \frac{\mathrm{L}_4}{2}\|\theta - \theta^\star\|^3\,.$$

Integrating (54) with respect to $\pi_\gamma$, we get

$$\mathrm{H}^\star(\bar{\theta}_\gamma - \theta^\star) + (1/2)\{\nabla^3 f(\theta^\star)\}\left[\int_{\mathbb{R}^d}(\theta - \theta^\star)^{\otimes 2}\pi_\gamma(\mathrm{d}\theta)\right] = -\int_{\mathbb{R}^d} G(\theta)\pi_\gamma(\mathrm{d}\theta)\,. \tag{55}$$

Moreover, using **C**1(6), we have

$$\|\int_{\mathbb{R}^d} G(\theta)\pi_\gamma(\mathrm{d}\theta)\| \leq \gamma^{3/2}\frac{\mathrm{L}\,\mathrm{D}_{\mathrm{last},6}^{3/2}\tau_6^3}{2\mu^{3/2}}\,. \tag{56}$$

Now we provide an explicit expression for the covariance matrix

$$\bar{\Sigma}_\gamma = \int_{\mathbb{R}^d}(\theta - \theta^\star)^{\otimes 2}\pi_\gamma(\mathrm{d}\theta)\,. \tag{57}$$

Using the recurrence (11), we obtain that

$$\theta_1 - \theta^\star = (\mathrm{I} - \gamma\,\mathrm{H}^\star)(\theta_0 - \theta^\star) - \gamma\varepsilon_1(\theta_0) - \gamma\eta(\theta_0)\,,$$

where the function $\eta(\cdot)$ is defined in (28). Hence, taking second moment w.r.t. $\pi_\gamma$ from both sides, we get that

$$\bar{\Sigma}_\gamma = (\mathrm{I} - \gamma\,\mathrm{H}^\star)\bar{\Sigma}_\gamma(\mathrm{I} - \gamma\,\mathrm{H}^\star) + \gamma^2\int_{\mathbb{R}^d}\mathcal{C}(\theta)\pi_\gamma(\mathrm{d}\theta) + \gamma^2\int_{\mathbb{R}^d}\{\eta(\theta)\}^{\otimes 2}\pi_\gamma(\mathrm{d}\theta)$$
$$- \gamma\int_{\mathbb{R}^d}\left[(\mathrm{I} - \gamma\,\mathrm{H}^\star)(\theta - \theta^\star)\{\eta(\theta)\}^\top + \eta(\theta)(\theta - \theta^\star)^\top(\mathrm{I} - \gamma\,\mathrm{H}^\star)\right]\pi_\gamma(\mathrm{d}\theta)\,. \tag{58}$$

In the above equation $\mathcal{C}(\theta)$ is defined in (50), and we additionally used that $\mathsf{E}\left[\varepsilon_1(\theta_0)|\mathcal{F}_0\right] = 0$. Using Taylor's expansion with integral remainder together with **A**2 and **C**1(6),

$$\gamma^2\|\int_{\mathbb{R}^d}\{\eta(\theta)\}^{\otimes 2}\pi_\gamma(\mathrm{d}\theta)\|_F \leq \gamma^4\frac{\mathrm{L}^2\,\mathrm{D}_{\mathrm{last},4}^2\tau_4^4}{4\mu^2}\,,$$

$$\gamma\|\int_{\mathbb{R}^d}\left[(\mathrm{I} - \gamma\,\mathrm{H}^\star)(\theta - \theta^\star)\{\eta(\theta)\}^\top + \eta(\theta)(\theta - \theta^\star)^\top(\mathrm{I} - \gamma\,\mathrm{H}^\star)\right]\pi_\gamma(\mathrm{d}\theta)\|_F \leq \gamma^{5/2}\frac{\mathrm{L}\,\mathrm{D}_{\mathrm{last},6}^{3/2}\tau_6^3}{2\mu^{3/2}}$$

Moreover, (50) together with **C**1(6) imply that

$$\int_{\mathbb{R}^d}\mathcal{C}(\theta)\pi_\gamma(\mathrm{d}\theta) = \mathcal{C}(\theta^\star) + B\gamma^{1/2}\,,$$

where $B \in \mathbb{R}^{d \times d}$ satisfies $\|B\| \leq \mathsf{C}_2$. Using (58) together with **C**1(6), we obtain that $\bar{\Sigma}_\gamma$ is a solution to the matrix equation

$$\mathrm{H}^\star\bar{\Sigma}_\gamma + \bar{\Sigma}_\gamma\,\mathrm{H}^\star - \gamma\,\mathrm{H}^\star\bar{\Sigma}_\gamma\,\mathrm{H}^\star = \gamma\mathcal{C}(\theta^\star) + B'\gamma^{3/2}\,, \tag{59}$$

where

$$\|B'\|_F \leq \mathsf{C}_2 + \frac{\mathrm{L}\,\mathrm{D}_{\mathrm{last},6}^{3/2}\tau_6^3}{2\mu^{3/2}} + \frac{\mathrm{L}^{1/2}\,\mathrm{D}_{\mathrm{last},4}^2\tau_4^4}{4\mu^2\mathsf{C}_{\mathrm{step},6}^{3/2}}\,. \tag{60}$$

The matrix equation (59) can be written using vectorization operation as

$$\text{vec}\left(\bar{\Sigma}_\gamma\right) = \gamma(\text{H}^\star \otimes \text{I} + \text{I} \otimes \text{H}^\star - \gamma \text{H}^\star \otimes \text{H}^\star)^{-1} \text{vec}\left(\mathcal{C}(\theta^\star)\right)$$
$$+ \gamma^{3/2}(\text{H}^\star \otimes \text{I} + \text{I} \otimes \text{H}^\star - \gamma \text{H}^\star \otimes \text{H}^\star)^{-1} \text{vec}\left(B'\right).$$

Applying Lemma 13(c), we obtain that

$$(\text{H}^\star \otimes \text{I} + \text{I} \otimes \text{H}^\star - \gamma \text{H}^\star \otimes \text{H}^\star)^{-1} = (\text{H}^\star \otimes \text{I} + \text{I} \otimes \text{H}^\star)^{-1} + D,$$

where $D \in \mathbb{R}^{d^2 \times d^2}$ is a matrix which satisfies

$$\|D\| \leq \gamma \text{L}/\mu.$$

Thus,

$$\text{vec}\left(\bar{\Sigma}_\gamma\right) = \gamma(\text{H}^\star \otimes \text{I} + \text{I} \otimes \text{H}^\star)^{-1} \text{vec}\left(\mathcal{C}(\theta^\star)\right) + \gamma^{3/2}(D/\sqrt{\gamma}) \text{vec}\left(\mathcal{C}(\theta^\star)\right)$$
$$+ \gamma^{3/2}(\text{H}^\star \otimes \text{I} + \text{I} \otimes \text{H}^\star - \gamma \text{H}^\star \otimes \text{H}^\star)^{-1} \text{vec}\left(B'\right).$$

We define the matrix $B_2$ such that

$$\text{vec}\left(B_2\right) = (D/\sqrt{\gamma}) \text{vec}\left(\mathcal{C}(\theta^\star)\right) + (\text{H}^\star \otimes \text{I} + \text{I} \otimes \text{H}^\star - \gamma \text{H}^\star \otimes \text{H}^\star)^{-1} \text{vec}\left(B'\right) \qquad (61)$$

Hence, using **A**3, (60), and Lemma 13, we get

$$\begin{aligned}
\|B_2\| &\leq \|B_2\|_F = \|\text{vec}\left(B_2\right)\| \\
&\leq \|D/\sqrt{\gamma}\| \|\text{vec}\left(\mathcal{C}(\theta^\star)\right)\| + \|(\text{H}^\star \otimes \text{I} + \text{I} \otimes \text{H}^\star - \gamma \text{H}^\star \otimes \text{H}^\star)^{-1}\| \|B'\|_F \\
&\leq \frac{\sqrt{\gamma} \text{L} \tau_2^2}{\mu} + \frac{1}{2}\left(\text{C}_2 + \frac{\text{L} \text{D}_{\text{last},6}^{3/2} \tau_6^3}{2\mu^{3/2}} + \frac{L^{1/2} \text{D}_{\text{last},4}^2 \tau_4^4}{4\mu^2 \text{C}_{\text{step},6}^{3/2}}\right) \\
&\leq \frac{\sqrt{\text{L}} \tau_2^2}{\sqrt{\text{C}_{\text{step},6}}\mu} + \frac{1}{2}\left(\text{C}_2 + \frac{\text{L} \text{D}_{\text{last},6}^{3/2} \tau_6^3}{2\mu^{3/2}} + \frac{L^{1/2} \text{D}_{\text{last},4}^2 \tau_4^4}{4\mu^2 \text{C}_{\text{step},6}^{3/2}}\right),
\end{aligned}$$

where in the last inequality we use that $\gamma \leq 1/(\text{L} \text{C}_{\text{step},6})$. Combining the above bounds in (55), we arrive at the expansion formula (51). $\qquad \square$

**Lemma 13.** *Assume **A**1 and **A**2. Then for any $\gamma \in (0, 1/(\text{L} \text{C}_{\text{step},6})]$ it holds*

(a) *All eigenvalues $\tilde{\lambda}_i$, $i \in \{1, \ldots, d^2\}$ of the matrix $\text{H}^\star \otimes \text{I} + \text{I} \otimes \text{H}^\star - \gamma \text{H}^\star \otimes \text{H}^\star$ satisfy*

$$2\mu(1 - \gamma \text{L}/2) \leq \tilde{\lambda}_i \leq 2\text{L}(1 - \gamma\mu/2);$$

(b) $\|(\text{H}^\star \otimes \text{I} + \text{I} \otimes \text{H}^\star - \gamma \text{H}^\star \otimes \text{H}^\star)^{-1}\| \leq 1/2;$

(c) *In addition,*

$$(\text{H}^\star \otimes \text{I} + \text{I} \otimes \text{H}^\star - \gamma \text{H}^\star \otimes \text{H}^\star)^{-1} = (\text{H}^\star \otimes \text{I} + \text{I} \otimes \text{H}^\star)^{-1} + D \text{ where } \|D\| \leq \gamma \text{L}/\mu.$$

*Proof.* Assumption **A** 1 guarantees that the symmetric matrix $\text{H}^\star$ is positive-definite. Let $u_1, \ldots, u_d \in \mathbb{R}^d$ and $\lambda_1 \geq \lambda_2 \geq \ldots \geq \lambda_d \geq \mu > 0$ be its eigenvectors and eigenvalues, respectively. Then we notice that

$$\text{H}^\star \otimes \text{I} + \text{I} \otimes \text{H}^\star - \gamma \text{H}^\star \otimes \text{H}^\star = \text{H}^\star \otimes (\text{I} - (\gamma/2) \text{H}^\star) + (\text{I} - (\gamma/2) \text{H}^\star) \otimes \text{H}^\star.$$

Hence, the latter operator is also diagonalizable in the orthogonal basis $u_i \otimes u_j \in \mathbb{R}^{d^2}$ with the respective eigenvalues being equal to $\lambda_i(1 - (\gamma/2)\lambda_j) + \lambda_j(1 - (\gamma/2)\lambda_i)$. Hence, we obtain the first part of lemma (a). To prove (b) it remains to note that for $\gamma \leq 1/\text{L}$ it holds $(2\mu(1 - \gamma \text{L}/2))^{-1} \leq 1/2$. Set now

$$\begin{aligned}
S &= \text{H}^\star \otimes \text{I} + \text{I} \otimes \text{H}^\star \in \mathbb{R}^{d^2 \times d^2} \\
R &= \text{H}^\star \otimes \text{H}^\star \in \mathbb{R}^{d^2 \times d^2}.
\end{aligned} \qquad (62)$$

Then it is easy to observe that

$$(S - \gamma R)^{-1} = S^{-1} + S^{-1} \sum_{k=1}^{\infty} \gamma^k (RS^{-1})^k \,,$$

provided that $\gamma \|RS^{-1}\| < 1$. Sice $R$ and $S$ are diagonalizable in the same orthogonal basis $\{u_i \otimes u_j\}_{1 \le i,j \le d}$ with the eigenvalues $\lambda_i \lambda_j$ and $\lambda_i + \lambda_j$, respectively, the condition $\gamma \|RS^{-1}\| < 1$ holds provided that $\gamma < 2/\mathrm{L}$. Hence, for $\gamma \le 1/\mathrm{L}$, it holds that

$$(\mathrm{H}^\star \otimes \mathrm{I} + \mathrm{I} \otimes \mathrm{H}^\star - \gamma \, \mathrm{H}^\star \otimes \mathrm{H}^\star)^{-1} = (\mathrm{H}^\star \otimes \mathrm{I} + \mathrm{I} \otimes \mathrm{H}^\star)^{-1} + D \,,$$

where $D \in \mathbb{R}^{d^2 \times d^2}$ satisfies

$$\|D\| \le 2\gamma \|S^{-1}\| \|RS^{-1}\| \le \frac{\gamma \, \mathrm{L}}{\mu} \,.$$

$\square$

We now state an auxiliary lemma about the function $\mathcal{C}(\theta)$ from (50).

**Lemma 14.** *Assume* **A**1*,* **A**2*,* **A**3*(2), and* **C**1*(2). Then, for any $\gamma \in (0, 1/(\mathrm{L}\,\mathsf{C}_{\mathrm{step},2})]$, it holds*

$$\| \int_{\mathbb{R}^d} \mathcal{C}(\theta)\pi_\gamma(\mathrm{d}\theta) - \mathcal{C}(\theta^\star)\|_F \le \mathsf{C}_2 \gamma^{1/2} \,,$$

*where the constant $\mathsf{C}_2$ is given by*

$$\mathsf{C}_2 = \left( \frac{\mathrm{L}^2 \, \mathsf{D}_{\mathrm{last},2}}{\mu^{3/2}} + \frac{\mathrm{L}\,\sqrt{\mathsf{D}_{\mathrm{last},2}}}{\sqrt{\mu}} \right) \tau_2^2 \,. \tag{63}$$

*Proof.* Recall that

$$\varepsilon_1(\theta) = \nabla F(\theta, \xi_1) - \nabla f(\theta) \,.$$

Hence, using the definition of $\mathcal{C}(\theta)$ in (50), we get, with $\theta \in \mathbb{R}^d$, that

$$\mathcal{C}(\theta) - \mathcal{C}(\theta^\star) = \mathsf{E}[(\varepsilon_1(\theta) - \varepsilon_1(\theta^\star))(\varepsilon_1(\theta) - \varepsilon_1(\theta^\star))^T] + \mathsf{E}[\varepsilon_1(\theta^\star)(\varepsilon_1(\theta) - \varepsilon_1(\theta^\star))^T]$$
$$+ \mathsf{E}[(\varepsilon_1(\theta) - \varepsilon_1(\theta^\star))\varepsilon_1(\theta^\star)^T].$$

Using **A**3(2), we obtain

$$\mathsf{E}[\|\varepsilon_1(\theta) - \varepsilon_1(\theta^\star)\|^2] \le \mathrm{L}\langle \nabla f(\theta) - \nabla f(\theta^\star), \theta - \theta^\star \rangle - \|\nabla f(\theta) - \nabla f(\theta^\star)\|^2 \le \mathrm{L}^2 \|\theta - \theta^\star\|^2.$$

Hence, combining the previous inequalities and using Hölder's inequality, we obtain for any $\theta \in \mathbb{R}^d$, that

$$\|\mathcal{C}(\theta) - \mathcal{C}(\theta^\star)\|_F \le \mathrm{L}^2 \|\theta - \theta^\star\|^2 + \tau_2 \mathrm{L} \|\theta - \theta^\star\|.$$

Applying now **C**1(2), we obtain

$$\| \int_{\mathbb{R}^d} \mathcal{C}(\theta)\pi_\gamma(\mathrm{d}\theta) - \mathcal{C}(\theta^\star)\|_F \le \int_{\mathbb{R}^d} \|\mathcal{C}(\theta) - \mathcal{C}(\theta^\star)\|_F \pi_\gamma(\mathrm{d}\theta) \le \mathrm{L}^2 \frac{\mathsf{D}_{\mathrm{last},2}\gamma\tau_2^2}{\mu} + \tau_2 \mathrm{L} \sqrt{\frac{\mathsf{D}_{\mathrm{last},2}\gamma\tau_2^2}{\mu}}.$$

We conclude the proof by using the fact that $\gamma\mu \le 1$. $\square$

Now we prove (24). We use synchronous coupling construction defined by the pair of recursions:

$$\theta_{k+1} = \theta_k - \gamma \nabla F(\theta_k, \xi_{k+1}), \quad \theta_0 \sim \nu$$
$$\tilde{\theta}_{k+1} = \tilde{\theta}_k - \gamma \nabla F(\tilde{\theta}_k, \xi_{k+1}), \quad \tilde{\theta}_0 \sim \pi_\gamma \,.$$

Recall that the corresponding coupling kernel is denoted as $\mathrm{K}_\gamma(\cdot, \cdot)$. Then we obtain

$$\mathsf{E}_\nu[\bar{\theta}_n] - \theta^\star = n^{-1} \sum_{k=n+1}^{2n} \mathsf{E}^{\mathrm{K}_\gamma}_{\nu,\pi_\gamma}[\theta_k - \tilde{\theta}_k] + n^{-1} \sum_{k=n+1}^{2n} \mathsf{E}_{\pi_\gamma}[\tilde{\theta}_k - \theta^\star]$$

$$= n^{-1} \sum_{k=n+1}^{2n} \mathsf{E}^{\mathrm{K}_\gamma}_{\nu,\pi_\gamma}[\theta_k - \tilde{\theta}_k] + (\bar{\theta}_\gamma - \theta^\star) \,.$$

Using (48) and **C**1(2), we obtain

$$
\begin{aligned}
\|\mathsf{E}^K_{\nu,\pi_\gamma}[\theta_k - \tilde{\theta}_k]\| &\leq (1-\gamma\mu)^{k/2}\{\mathsf{E}^{\mathrm{K}_\gamma}_{\nu,\pi_\gamma}\|\theta_0 - \tilde{\theta}_0\|^2\}^{1/2} \\
&\leq (1-\gamma\mu)^{k/2}(\mathsf{E}^{1/2}_\nu\big[\|\theta_0 - \theta^\star\|^2\big] + \frac{\sqrt{2\gamma}\tau_2}{\sqrt{\mu}}).
\end{aligned}
$$

Summing the above bounds for $k$ from $n+1$ to $2n$, we obtain (24).

## A.3 Proof of Proposition 5

Note that

$$
\mathsf{E}_\nu[\bar{\theta}^{(RR)}_n - \theta^\star] = 2\mathsf{E}_\nu[\bar{\theta}^\gamma_n - \theta^\star] - \mathsf{E}_\nu[\bar{\theta}^{2\gamma}_n - \theta^\star].
$$

Applying (24), we obtain

$$
\|\mathsf{E}_\nu[\bar{\theta}^{(RR)}_n - \theta^\star]\| \leq (\frac{L}{\mu}\mathsf{C}_1 + \frac{L\mathsf{D}^{3/2}_{\mathrm{last},6}\tau^3_6}{2\mu^{5/2}})\gamma^{3/2} + \mathcal{R}_3(\theta_0 - \theta^\star, \gamma, n), \tag{64}
$$

where

$$
\|\mathcal{R}_3(\theta_0 - \theta^\star, \gamma, n)\| \lesssim \frac{(1-\gamma\mu)^{(n+1)/2}}{n\gamma\mu}(\mathsf{E}^{1/2}_\nu\big[\|\theta_0 - \theta^\star\|^2\big] + \frac{\sqrt{\gamma}\tau_2}{\sqrt{\mu}}), \tag{65}
$$

and the statement follows.

## B Proof of Theorem 3

**Theorem 15** (Version of Theorem 3 with explicit constants). *Assume **A**1, **A**2, **A**3(6), and **C**1(6). Then for any $\gamma \in (0, 1/(\mathrm{L}\,\mathsf{C}_{\mathrm{step},6})]$, $n \in \mathbb{N}$, and initial distribution $\nu$ on $\mathbb{R}^d$, the sequence of Polyak-Ruppert estimates (14) satisfies*

$$
\mathsf{E}^{1/2}_\nu[\|\,\mathrm{H}^\star(\bar{\theta}^{(\gamma)}_n - \theta^\star)\|^2] \leq \frac{\sqrt{\mathrm{Tr}\,\Sigma^\star_\varepsilon}}{\sqrt{n}} + \frac{\mathsf{C}_2}{\gamma^{1/2}n} + \mathsf{C}_3\gamma + \frac{\mathsf{C}_4\gamma^{1/2}}{n^{1/2}} + \mathcal{R}_2(n, \gamma, \|\theta_0 - \theta^\star\|),
$$

*where we have set*

$$
\mathsf{C}_2 = c_0\mathsf{D}^{1/2}_{\mathrm{last},2}\tau_2, \quad \mathsf{C}_3 = c_0\frac{\mathrm{L}\,\mathsf{D}_{\mathrm{last},4}\tau^2_4}{2\mu}, \quad \mathsf{C}_4 = c_0\,\mathrm{L}\,\mathsf{D}^{1/2}_{\mathrm{last},2}\tau_2. \tag{66}
$$

*and the remainder term $\mathcal{R}_2(n, \gamma, \|\theta_0 - \theta^\star\|)$ is given by*

$$
\begin{aligned}
\mathcal{R}_2(n, \gamma, \|\theta_0 - \theta^\star\|) = \frac{c_0\,\mathrm{L}(1-\gamma\mu)^{(n+1)/2}}{\gamma\mu n}&\mathsf{E}^{1/2}_\nu\big[\|\theta_0 - \theta^\star\|^2\big] \\
&+ \frac{\mathrm{L}\,c_0(1-\gamma\mu)^{n+1}}{2n\gamma\mu}\mathsf{E}^{1/2}_\nu\big[\|\theta_0 - \theta^\star\|^4\big]. \quad (67)
\end{aligned}
$$

*Proof.* Throughout the proof we omit upper index $(\gamma)$ both for the elements of the sequence $\{\theta^{(\gamma)}_k\}_{k\in\mathbb{N}}$ and Polyak-Ruppert averaged estimates $\bar{\theta}^{(\gamma)}_n$. Instead, we write simply $\theta_k$ and $\bar{\theta}_n$, respectively. Summing the recurrence (30), we obtain that

$$
\mathrm{H}^\star(\bar{\theta}_n - \theta^\star) = \frac{\theta_{n+1} - \theta^\star}{\gamma n} - \frac{\theta_{2n+1} - \theta^\star}{\gamma n} - \frac{1}{n}\sum^{2n}_{k=n+1}\varepsilon_{k+1}(\theta_k) - \frac{1}{n}\sum^{2n}_{k=n+1}\eta(\theta_k). \tag{68}
$$

Applying the 3-rd order Taylor expansion with integral remainder, we get that

$$
\nabla f(\theta_k) = \mathrm{H}^\star(\theta_k - \theta^\star) + \left(\int^1_0 t\nabla^3 f(t\theta^\star + (1-t)\theta_k)\,dt\right)(\theta_k - \theta^\star)^{\otimes 2},
$$

where $\nabla^3 f(\cdot) \in \mathbb{R}^{d\times d\times d}$. Using **A**2, we thus obtain that

$$
\|\eta(\theta_k)\| \leq \frac{1}{2}\mathrm{L}_3\|\theta_k - \theta^\star\|^2.
$$

Applying Minkowski's inequality to the decomposition (33) and to the last term therein, we get

$$\mathsf{E}_\nu^{1/2}[\| \, \mathrm{H}^\star(\bar\theta_n - \theta^\star)\|^2] \leq \frac{\mathsf{E}_\nu^{1/2}[\|\theta_{n+1} - \theta^\star\|^2]}{\gamma n} + \frac{\mathsf{E}_\nu^{1/2}[\|\theta_{2n+1} - \theta^\star\|^2]}{\gamma n} + \frac{1}{n}\mathsf{E}_\nu^{1/2}\Big[\|\sum_{k=n+1}^{2n}\varepsilon_{k+1}(\theta_k)\|^2\Big]$$

$$+ \frac{\mathrm{L}_3}{2n}\sum_{k=n+1}^{2n}\mathsf{E}_\nu^{1/2}\big[\|\theta_k - \theta^\star\|^4\big].$$

Applying **C**1(2), we obtain that for $\gamma \in (0; 1/(\mathrm{L}\,\mathsf{C}_{\mathrm{step},2})]$ it holds that

$$\mathsf{E}_\nu\|\theta_k - \theta^\star\|^2 \lesssim (1 - \gamma\mu)^k \mathsf{E}_\nu\big[\|\theta_0 - \theta^\star\|^2\big] + \frac{\mathsf{D}_{\mathrm{last},2}\gamma\tau_2^2}{\mu}. \tag{69}$$

Moreover, from $\gamma \in (0; 1/(\mathrm{L}\,\mathsf{C}_{\mathrm{step},4})]$ it holds that

$$\mathsf{E}_\nu^{1/2}\|\theta_k - \theta^\star\|^4 \lesssim (1 - \gamma\mu)^k \mathsf{E}_\nu^{1/2}\big[\|\theta_0 - \theta^\star\|^4\big] + \frac{\mathsf{D}_{\mathrm{last},4}\gamma\tau_4^2}{\mu}. \tag{70}$$

Combining Lemma 16 with previous inequalities, we obtain

$$\mathsf{E}_\nu^{1/2}[\| \, \mathrm{H}^\star(\bar\theta_n - \theta^\star)\|^2] \lesssim \frac{\sqrt{\mathrm{Tr}\,\Sigma_\varepsilon^\star}}{\sqrt{n}} + \frac{\mathsf{D}_{\mathrm{last},2}^{1/2}\tau_2}{\gamma^{1/2}n} + \frac{\mathrm{L}\,\mathsf{D}_{\mathrm{last},4}\gamma\tau_4^2}{2\mu} + \frac{\mathrm{L}\,\mathsf{D}_{\mathrm{last},2}^{1/2}\gamma^{1/2}\tau_2}{\mu^{1/2}n^{1/2}}$$

$$+ \frac{(1 - \gamma\mu)^{(n+1)/2}}{\gamma n}\Big(\frac{\mathrm{L}}{\mu} + 1\Big)\mathsf{E}_\nu^{1/2}\big[\|\theta_0 - \theta^\star\|^2\big] + \frac{\mathrm{L}(1 - \gamma\mu)^{n+1}}{n\gamma\mu}\mathsf{E}_\nu^{1/2}\big[\|\theta_0 - \theta^\star\|^4\big],$$

and the result follows. $\qquad\square$

Below we provide an auxiliary lemma used in the proof of Theorem 3.

**Lemma 16.** *Assume **A**1, **A**2, **A**3(2), and **C**1(2). Then for any $\gamma \in (0; 1/(\mathrm{L}\,\mathsf{C}_{\mathrm{step},2})]$ and any $n \in \mathbb{N}$, it holds*

$$\mathsf{E}_\nu^{1/2}\Big[\|\sum_{k=n+1}^{2n}\{\varepsilon_{k+1}(\theta_k) - \varepsilon_{k+1}(\theta^\star)\}\|^2\Big] \lesssim \frac{\mathrm{L}\,\mathsf{D}_{\mathrm{last},2}^{1/2}\sqrt{\gamma n}\tau_2}{\mu^{1/2}} + \frac{\mathrm{L}(1 - \gamma\mu)^{(n+1)/2}}{\gamma\mu}\mathsf{E}_\nu^{1/2}\big[\|\theta_0 - \theta^\star\|^2\big]. \tag{71}$$

*Moreover, let $p \geq 2$, and assume **A**1, **A**2, **A**3(p), and **C**1(p). Then for any $\gamma \in (0; 1/(\mathrm{L}\,\mathsf{C}_{\mathrm{step},p})]$ and $n \in \mathbb{N}$ it holds that*

$$\mathsf{E}_\nu^{1/p}\Big[\|\sum_{k=n+1}^{2n}\{\varepsilon_{k+1}(\theta_k) - \varepsilon_{k+1}(\theta^\star)\}\|^p\Big] \lesssim \frac{\mathrm{L}\,\mathsf{D}_{\mathrm{last},p}^{1/2}\sqrt{\gamma n}p\tau_p}{\mu^{1/2}}$$

$$+ \frac{\mathrm{L}\,p(1 - \gamma\mu)^{(n+1)/2}}{\mu^{1/2}\gamma^{1/2}}\mathsf{E}_\nu^{1/p}[\|\theta_0 - \theta^\star\|^p]. \tag{72}$$

*Proof.* Since $\{\varepsilon_{k+1}(\theta_k) - \varepsilon_{k+1}(\theta^\star)\}$ is a martingale-difference sequence with respect to $\mathcal{F}_k$, we have

$$\mathsf{E}_\nu\Big[\|\sum_{k=n+1}^{2n}\{\varepsilon_{k+1}(\theta_k) - \varepsilon_{k+1}(\theta^\star)\}\|^2\Big] = \sum_{k=n+1}^{2n}\mathsf{E}_\nu\big[\|\{\varepsilon_{k+1}(\theta_k) - \varepsilon_{k+1}(\theta^\star)\}\|^2\big].$$

where $\varepsilon_{k+1}(\theta^\star) = \nabla F(\theta^\star, \xi_{k+1})$ uses the same noise variable $\xi_{k+1}$ as $F(\theta_k, \xi_{k+1})$. Note that

$$\mathsf{E}_\nu[\|\varepsilon_{k+1}(\theta_k) - \varepsilon_{k+1}(\theta^\star)\|^2] = \mathsf{E}_\nu[\|\nabla F(\theta_k, \xi_{k+1}) - \nabla F(\theta^\star, \xi_{k+1})\|^2$$

$$- 2\mathsf{E}_\nu\big[\langle\nabla F(\theta_k, \xi_{k+1}) - \nabla F(\theta^\star, \xi_{k+1}), \nabla f(\theta_k) - \nabla f(\theta^\star)\rangle\big] + \|\nabla f(\theta_k) - \nabla f(\theta^\star)\|^2\big].$$

Using **A**2, **A**3(2), and taking conditional expectation with respect to $\mathcal{F}_k$, we obtain

$$\mathsf{E}_\nu[\|\varepsilon_{k+1}(\theta_k) - \varepsilon_{k+1}(\theta^\star)\|^2] \leq \mathsf{E}_\nu[\mathrm{L}\langle\nabla f(\theta_k) - \nabla f(\theta^\star), \theta_k - \theta^\star\rangle - \|\nabla f(\theta_k) - \nabla f(\theta^\star)\|^2]$$

$$\leq \mathrm{L}^2\,\mathsf{E}_\nu[\|\theta_k - \theta^\star\|^2].$$

Thus, we obtain that

$$\mathsf{E}_\nu[\|\sum_{k=n+1}^{2n} \{\varepsilon_{k+1}(\theta_k) - \varepsilon_{k+1}(\theta^\star)\}\|^2] \leq \mathsf{L}^2 \sum_{k=n+1}^{2n} \mathsf{E}_\nu[\|\theta_k - \theta^\star\|^2],$$

and the statement (71) follows from the assumption **C**1(2). In order to prove (72), we apply Burkholder's inequality Osekowski (2012, Theorem 8.6) and obtain

$$\mathsf{E}_\nu^{1/p}[\|\sum_{k=n+1}^{2n} \{\varepsilon_{k+1}(\theta_k) - \varepsilon_{k+1}(\theta^\star)\}\|^p] \leq p\mathsf{E}_\nu^{1/p}\big[\big(\sum_{k=n+1}^{2n} \|\varepsilon_{k+1}(\theta_k) - \varepsilon_{k+1}(\theta^\star)\|^2\big)^{p/2}\big]$$

$$\leq p\big(\sum_{k=n+1}^{2n} \mathsf{E}_\nu^{2/p}\big[\|\varepsilon_{k+1}(\theta_k) - \varepsilon_{k+1}(\theta^\star)\|^p\big]\big)^{1/2}$$

$$\lesssim p\,\mathsf{L}\big(\sum_{k=n+1}^{2n} \mathsf{E}_\nu^{2/p}\big[\|\theta_k - \theta^\star\|^p\big]\big)^{1/2}$$

$$\overset{(a)}{\lesssim} \frac{\mathsf{L}\,\mathsf{D}_{\text{last},p}^{1/2}\sqrt{\gamma n}p\tau_p}{\mu^{1/2}} + \frac{\mathsf{L}\,p(1-\gamma\mu)^{(n+1)/2}}{\mu^{1/2}\gamma^{1/2}}\mathsf{E}_\nu^{1/p}[\|\theta_0 - \theta^\star\|^p],$$

where in (a) we have additionally used **C**1(p). $\qquad\square$

## C   Proof of Theorem 6

Within this section we often use the definition of the function $\psi : \mathbb{R}^d \to \mathbb{R}^d$ from (38):

$$\psi(\theta) = (1/2)\nabla^3 f(\theta^*)(\theta - \theta^\star)^{\otimes 2} \tag{73}$$

**Theorem 17** (Version of Theorem 6 with explicit constants). *Assume **A**1, **A**2, **A**3(6), and **C**1(6). Then for any $\gamma \in (0, 1/(\mathsf{L}\,\mathsf{C}_{\text{step},6}) \wedge 2/(11\,\mathsf{L})]$, initial distribution $\nu$, and $n \in \mathbb{N}$, the Richardson-Romberg estimator $\bar{\theta}_n^{(RR)}$ defined in (37) satisfies*

$$\mathsf{E}_\nu^{1/2}[\|\,\mathsf{H}^\star(\bar{\theta}_n^{(RR)} - \theta^\star)\|^2] \leq \frac{\sqrt{\mathrm{Tr}\,\Sigma_\varepsilon^\star}}{n^{1/2}} + \frac{\mathsf{C}_{\text{RR},1}\gamma^{1/2}}{n^{1/2}} + \frac{\mathsf{C}_{\text{RR},2}}{\gamma^{1/2}n} + \mathsf{C}_{\text{RR},3}\gamma^{3/2} + \frac{\mathsf{C}_{\text{RR},4}\gamma}{n^{1/2}}$$

$$+ \mathcal{R}_4(n, \gamma, \|\theta_0 - \theta^\star\|),$$

*where we have set*

$$\mathsf{C}_{\text{RR},1} = \frac{c_0\mathsf{D}_{\text{last},4}\,\mathsf{L}\,\tau_4^2}{\mu^{3/2}} + \frac{c_0\,\mathsf{L}\,\mathsf{D}_{\text{last},2}^{1/2}\tau_2}{\mu^{1/2}}, \quad \mathsf{C}_{\text{RR},2} = \frac{c_0\mathsf{D}_{\text{last},2}^{1/2}\tau_2}{\mu^{1/2}}$$

$$\mathsf{C}_{\text{RR},3} = c_0\left(\frac{\mathsf{L}\,\mathsf{D}_{\text{last},6}^{3/2}\tau_6^3}{\mu^{3/2}} + \mathsf{C}_1\right), \quad \mathsf{C}_{\text{RR},4} = \frac{c_0\mathsf{D}_{\text{last},4}\,\mathsf{L}\,\tau_4^2}{\mu}, \tag{74}$$

$\mathsf{C}_1$ *is defined in (52), and the remainder term $\mathcal{R}_4(n, \gamma, \|\theta_0 - \theta^\star\|)$ is given by*

$$\mathcal{R}_4(n, \gamma, \|\theta_0 - \theta^\star\|) = \frac{c_0\,\mathsf{L}(1-\gamma\mu)^{(n+1)/2}}{n\gamma\mu}$$

$$\times \left(\mathsf{E}_\nu^{1/2}[\|\theta_0 - \theta^\star\|^6] + \mathsf{E}_\nu^{1/2}[\|\theta_0 - \theta^\star\|^4] + \mathsf{E}_\nu^{1/2}[\|\theta_0 - \theta^\star\|^2] + \frac{\mathsf{D}_{\text{last},4}\gamma\tau_4^2}{\mu}\right). \tag{75}$$

*Proof.* Using the recursion (30), we obtain that

$$\mathsf{H}^\star(\bar{\theta}_n^{(RR)} - \theta^\star) = \frac{2(\theta_{n+1}^{(\gamma)} - \theta^\star)}{\gamma n} - \frac{2(\theta_{2n}^{(\gamma)} - \theta^\star)}{\gamma n} - \frac{\theta_{n+1}^{(2\gamma)} - \theta^\star}{2\gamma n} + \frac{\theta_{2n}^{(2\gamma)} - \theta^\star}{2\gamma n}$$

$$- \frac{1}{n}\sum_{k=n+1}^{2n} [2\varepsilon_{k+1}(\theta_k^{(\gamma)}) - \varepsilon_{k+1}(\theta_k^{(2\gamma)})] - \frac{1}{n}\sum_{k=n+1}^{2n} [2\eta(\theta_k^{(\gamma)}) - \eta(\theta_k^{(2\gamma)})]. \tag{76}$$

Therefore, applying Minkowski's inequality to the decomposition (76), we obtain for any initial distribution $\nu$ that

$$
\mathsf{E}_\nu^{1/2}[\|\,\mathrm{H}^\star(\bar\theta_n^{(RR)} - \theta^\star)\|^2] \leq \underbrace{\frac{1}{n}\mathsf{E}_\nu^{1/2}[\|\sum_{k=n+1}^{2n}\varepsilon_{k+1}(\theta^\star)\|^2]}_{T_1} + \underbrace{\frac{2}{\gamma n}\mathsf{E}_\nu^{1/2}[\|\theta_{n+1}^{(\gamma)} - \theta^\star\|^2] + \frac{2}{\gamma n}\mathsf{E}_\nu^{1/2}[\|\theta_{2n+1}^{(\gamma)} - \theta^\star\|^2]}_{T_2}
$$

$$
+ \underbrace{\frac{1}{2\gamma n}\mathsf{E}_\nu^{1/2}[\|\theta_{n+1}^{(2\gamma)} - \theta^\star\|^2] + \frac{1}{2\gamma n}\mathsf{E}_\nu^{1/2}[\|\theta_{2n+1}^{(2\gamma)} - \theta^\star\|^2]}_{T_3}
$$

$$
+ \underbrace{\frac{2}{n}\mathsf{E}_\nu^{1/2}[\|\sum_{k=n+1}^{2n}\varepsilon_{k+1}(\theta_k^{(\gamma)}) - \varepsilon_{k+1}(\theta^\star)\|^2]}_{T_4}
$$

$$
+ \underbrace{\frac{1}{n}\mathsf{E}_\nu^{1/2}[\|\sum_{k=n+1}^{2n}\varepsilon_{k+1}(\theta_k^{(2\gamma)}) - \varepsilon_{k+1}(\theta^\star)\|^2]}_{T_5} + \underbrace{\|2\pi_\gamma(\psi) - \pi_{2\gamma}(\psi)\|}_{T_6}
$$

$$
+ \underbrace{\frac{2}{n}\mathsf{E}_\nu^{1/2}[\|\sum_{k=n+1}^{2n}\eta(\theta_k^{(\gamma)}) - \pi_\gamma(\psi)\|^2] + \frac{1}{n}\mathsf{E}_\nu^{1/2}[\|\sum_{k=n+1}^{2n}\eta(\theta_k^{(2\gamma)}) - \pi_{2\gamma}(\psi)\|^2]}_{T_7} .
$$

Now we upper bound the terms in the right-hand side of the above bound separately. First, we note that

$$
T_1 = \frac{\sqrt{\mathrm{Tr}\,\Sigma_\varepsilon^\star}}{\sqrt{n}} .
$$

Using **C**1(2), we get

$$
T_2 + T_3 \lesssim \frac{(1 - \gamma\mu)^{(n+1)/2}}{\gamma n}\mathsf{E}_\nu^{1/2}[\|\theta_0 - \theta^\star\|^2] + \frac{\mathsf{D}_{\mathrm{last},2}^{1/2}\tau_2}{\mu^{1/2}\gamma^{1/2}n} .
$$

Applying Lemma 16, we get

$$
T_4 + T_5 \lesssim \frac{\mathrm{L}\,\mathsf{D}_{\mathrm{last},2}^{1/2}\gamma^{1/2}\tau_2}{\mu^{1/2}n^{1/2}} + \frac{\mathrm{L}(1 - \gamma\mu)^{(n+1)/2}}{\mu\gamma n}\mathsf{E}_\nu^{1/2}[\|\theta_0 - \theta^\star\|^2] .
$$

Now we proceed with the term $T_6$. Applying the recurrence (11), we obtain that

$$
\theta_1^{(\gamma)} - \theta^\star = (\mathrm{I} - \gamma\,\mathrm{H}^\star)(\theta_0^{(\gamma)} - \theta^\star) - \gamma\varepsilon_1(\theta_0^{(\gamma)}) - \gamma\eta(\theta_0^{(\gamma)}) . \tag{77}
$$

Thus, taking expectation w.r.t. $\pi_\gamma$ in both sides above, we get

$$
\mathrm{H}^\star(\bar\theta_\gamma - \theta^\star) = \mathsf{E}_{\pi_\gamma}[\eta(\theta_0^{(\gamma)})] = \pi_\gamma(\psi) + \pi_\gamma(G) ,
$$

where $G(\theta)$ is defined in (38) and writes as

$$
G(\theta) = \frac{1}{2}\left(\int_0^1 t^2\nabla^4 f(t\theta^\star + (1-t)\theta)\,dt\right)(\theta - \theta^\star)^{\otimes 3} .
$$

Hence, applying **A**2 together with Proposition 2, we obtain that

$$
T_6 = \|2\pi_\gamma(\psi) - \pi_{2\gamma}(\psi)\| \lesssim \mathsf{C}_1\gamma^{3/2} . \tag{78}
$$

Finally, using Lemma 21, Lemma 20, and Lemma 18, we obtain that

$$
T_7 \lesssim \frac{\mathsf{D}_{\mathrm{last},4}\,\mathrm{L}\,\gamma\tau_4^2}{\mu n^{1/2}} + \frac{\mathsf{D}_{\mathrm{last},4}\,\mathrm{L}\,\gamma^{1/2}\tau_4^2}{\mu^{3/2}n^{1/2}} + \frac{\mathrm{L}\,\mathsf{D}_{\mathrm{last},6}^{3/2}\gamma^{3/2}\tau_6^3}{\mu^{3/2}}
$$

$$
+ \frac{\mathrm{L}(1 - \gamma\mu)^{(n+1)/2}}{n\gamma\mu}\left(\mathsf{E}_\nu^{1/2}[\|\theta_0 - \theta^\star\|^6] + \mathsf{E}_\nu^{1/2}[\|\theta_0 - \theta^\star\|^4] + \frac{\mathsf{D}_{\mathrm{last},4}\gamma\tau_4^2}{\mu}\right) .
$$

Combining the bounds above completes the proof. $\qquad\square$

Below we provide some auxiliary technical lemmas.

**Lemma 18.** *Assume A1, A2, A3(4), and C1(4). Then for any $\gamma \in (0; 1/(\mathsf{L}\,\mathsf{C}_{\mathrm{step},4})]$ and any $n \in \mathbb{N}$ it holds*

$$n^{-1}\mathsf{E}_{\pi_\gamma}^{1/2}\big[\|\sum_{k=n+1}^{2n}\{\psi(\theta_k)-\pi_\gamma(\psi)\}\|^2\big] \lesssim \frac{\mathsf{D}_{\mathrm{last},4}\,\mathsf{L}_3\,\gamma\tau_4^2}{\mu n^{1/2}} + \frac{\mathsf{D}_{\mathrm{last},4}\,\mathsf{L}_3\,\gamma^{1/2}\tau_4^2}{\mu^{3/2}n^{1/2}}\,. \tag{79}$$

*Proof.* Using the fact that $\pi_\gamma$ is a stationary distribution, we obtain that

$$\mathsf{E}_{\pi_\gamma}\big[\|\sum_{k=n+1}^{2n}\{\psi(\theta_k)-\pi_\gamma(\psi)\}\|^2\big] = n\mathsf{E}_{\pi_\gamma}[\|\psi(\theta_0)-\pi_\gamma(\psi)\|^2]$$

$$+ \sum_{k=1}^{n-1}(n-k)\mathsf{E}_{\pi_\gamma}[(\psi(\theta_0)-\pi_\gamma(\psi))^T(\psi(\theta_k)-\pi_\gamma(\psi))]$$

Using the Markov property, Cauchy–Schwartz inequality, Proposition 1, and Lemma 22, we obtain

$$\begin{aligned}
&\mathsf{E}_{\pi_\gamma}[(\psi(\theta_0)-\pi_\gamma(\psi))^T(\psi(\theta_k)-\pi_\gamma(\psi))] \\
&= \mathsf{E}_{\pi_\gamma}[(\psi(\theta_0)-\pi_\gamma(\psi))^T(\mathsf{Q}_\gamma^k\psi(\theta_0)-\pi_\gamma(\psi))] \\
&\overset{(a)}{\lesssim} (1/2)^{k/m(\gamma)}\,\mathsf{L}_3\,\mathsf{E}_{\pi_\gamma}\big[\|\psi(\theta_0)-\pi_\gamma(\psi)\|\int c(\theta_0,\vartheta)\mathrm{d}\pi_\gamma(\vartheta)\big]\,,
\end{aligned} \tag{80}$$

where in (a) we additionally used the fact that

$$\mathbf{W}_c(\delta_{\theta_0},\pi_\gamma) = \int c(\theta_0,\vartheta)\mathrm{d}\pi_\gamma(\vartheta)\,.$$

Using C1(4), we get

$$\mathsf{E}_{\pi_\gamma}[\|\psi(\theta_0)-\pi_\gamma\|^2] \le \mathsf{E}_{\pi_\gamma}[\|\psi(\theta_0)\|^2] \le \mathsf{L}_3^2\,\mathsf{E}_{\pi_\gamma}[\|\theta_0-\theta^\star\|^4] \le \frac{\mathsf{L}_3^2\,\mathsf{D}_{\mathrm{last},4}\gamma^2\tau_4^4}{\mu^2}\,, \tag{81}$$

and, using C1(2) and C1(4), we get

$$\int\int c^2(\theta_0,\vartheta)d\pi_\gamma(\vartheta)d\pi_\gamma(\theta_0) \tag{82}$$

$$\le \int\int\|\theta_0-\vartheta\|^2\left(\|\theta_0-\theta^\star\|+\|\vartheta-\theta^\star\|+\frac{2^{3/2}\gamma^{1/2}\tau_2}{\mu^{1/2}}\right)^2 d\pi_\gamma(\vartheta)d\pi_\gamma(\theta_0) \tag{83}$$

$$\lesssim \int\int(\|\theta_0-\theta^\star\|^4+\|\vartheta-\theta^\star\|^4)+\frac{\gamma\tau_2^2}{\mu}(\|\theta_0-\theta^\star\|^2+\|\vartheta-\theta^\star\|^2)d\pi_\gamma(\vartheta)d\pi_\gamma(\theta_0) \tag{84}$$

$$\lesssim \frac{\mathsf{D}_{\mathrm{last},4}\gamma^2\tau_4^4}{\mu^2} + \frac{\mathsf{D}_{\mathrm{last},2}\gamma^2\tau_2^4}{\mu^2} \lesssim \frac{\mathsf{D}_{\mathrm{last},4}\gamma^2\tau_4^4}{\mu^2}\,. \tag{85}$$

Using (81), (82), and Cauchy–Schwartz inequality for (80), we obtain

$$\mathsf{E}_{\pi_\gamma}[(\psi(\theta_0)-\pi_\gamma(\psi))^T(\psi(\theta_k)-\pi_\gamma(\psi))] \lesssim (1/2)^{k/m(\gamma)}\frac{\mathsf{L}_3\,\mathsf{D}_{\mathrm{last},4}\gamma^2\tau_4^4}{\mu^2}\,.$$

Combining the inequalities above and using that $m(\gamma) = \lceil 2\frac{\log 4}{\gamma\mu}\rceil \le \frac{2\log 4+1}{\gamma\mu}$, we get

$$\begin{aligned}
n^{-1}\mathsf{E}_{\pi_\gamma}^{1/2}\big[\|\sum_{k=n+1}^{2n}\{\psi(\theta_k)-\pi_\gamma(\psi)\}\|^2\big] &\le \left(\frac{\mathsf{D}_{\mathrm{last},4}\,\mathsf{L}_3^2\,\gamma^2\tau_4^4}{\mu^2 n} + \frac{\mathsf{D}_{\mathrm{last},4}m(\gamma)\,\mathsf{L}_3^2\,\gamma^2\tau_4^4}{\mu^2 n}\right)^{1/2} \\
&\lesssim \frac{\mathsf{D}_{\mathrm{last},4}\,\mathsf{L}_3\,\gamma\tau_4^2}{\mu n^{1/2}} + \frac{\mathsf{D}_{\mathrm{last},4}\,\mathsf{L}_3\,\gamma^{1/2}\tau_4^2}{\mu^{3/2}n^{1/2}}\,.
\end{aligned}$$

$\square$

**Lemma 19.** *Assume A1, A2, A3(4). Then for any $\gamma \in (0; \frac{2}{11\,\mathrm{L}}]$, and any $k \in \mathbb{N}$ it holds that*

$$\mathsf{E}[\|\theta_{k+1} - \tilde{\theta}_{k+1}\|^4 | \mathcal{F}_k] \leq (1 - \gamma\mu)^2 \|\theta_k - \tilde{\theta}_k\|^4. \tag{86}$$

*Moreover, let $p \geq 2$ and assume A1, A2, and A3(2p). Then for any $\gamma \in (0, \frac{p}{4 \cdot 3^p\,\mathrm{L}}]$ and any $k \in \mathbb{N}$ it holds that*

$$\mathsf{E}[\|\theta_{k+1} - \tilde{\theta}_{k+1}\|^{2p} | \mathcal{F}_k] \leq (1 - \gamma\mu)^p \|\theta_k - \tilde{\theta}_k\|^{2p}. \tag{87}$$

*Proof.* Recall that the sequences $\{\theta_k\}_{k\in\mathbb{N}}$ and $\{\tilde{\theta}_k\}_{k\in\mathbb{N}}$ are defined by the recurrences

$$\theta_{k+1} = \theta_k - \gamma\nabla F(\theta_k, \xi_{k+1}), \quad \theta_0 = \theta \in \mathbb{R}^d, \tag{88}$$

$$\tilde{\theta}_{k+1} = \tilde{\theta}_k - \gamma\nabla F(\tilde{\theta}_k, \xi_{k+1}), \quad \tilde{\theta}_0 = \tilde{\theta} \in \mathbb{R}^d. \tag{89}$$

Expanding the brackets, we obtain that

$$\|\theta_{k+1} - \tilde{\theta}_{k+1}\|^4 = \|\theta_k - \tilde{\theta}_k\|^4 + \gamma^4\|\nabla F(\theta_k, \xi_{k+1}) - \nabla F(\tilde{\theta}_k, \xi_{k+1})\|^4$$
$$+ 4\gamma^2\langle\nabla F(\theta_k, \xi_{k+1}) - \nabla F(\tilde{\theta}_k, \xi_{k+1}), \theta_k - \tilde{\theta}_k\rangle^2$$
$$+ 2\gamma^2\|\nabla F(\theta_k, \xi_{k+1}) - \nabla F(\tilde{\theta}_k, \xi_{k+1})\|^2\|\theta_k - \tilde{\theta}_k\|^2$$
$$- 4\gamma\langle\nabla F(\theta_k, \xi_{k+1}) - \nabla F(\tilde{\theta}_k, \xi_{k+1}), \theta_k - \tilde{\theta}_k\rangle\|\theta_k - \tilde{\theta}_k\|^2$$
$$- 4\gamma^3\langle\nabla F(\theta_k, \xi_{k+1}) - \nabla F(\tilde{\theta}_k, \xi_{k+1}), \theta_k - \tilde{\theta}_k\rangle\|\nabla F(\theta_k, \xi_{k+1}) - \nabla F(\tilde{\theta}_k, \xi_{k+1})\|^2$$

Using A3(4) and Cauchy–Schwartz inequality, we get

$$\mathsf{E}[\|\nabla F(\theta_k, \xi_{k+1}) - \nabla F(\tilde{\theta}_k, \xi_{k+1})\|^4 | \mathcal{F}_k] \leq \mathrm{L}^3\langle\nabla f(\theta_k) - \nabla f(\tilde{\theta}_k), \theta_k - \tilde{\theta}_k\rangle\|\theta_k - \tilde{\theta}_k\|^2,$$

$$\mathsf{E}[\langle\nabla F(\theta_k, \xi_{k+1}) - \nabla F(\tilde{\theta}_k, \xi_{k+1}), \theta_k - \tilde{\theta}_k\rangle^2 | \mathcal{F}_k] \leq \mathrm{L}\langle\nabla f(\theta_k) - \nabla f(\tilde{\theta}_k), \theta_k - \tilde{\theta}_k\rangle\|\theta_k - \tilde{\theta}_k\|^2,$$

$$\mathsf{E}[\|\nabla F(\theta_k, \xi_{k+1}) - \nabla F(\tilde{\theta}_k, \xi_{k+1})\|^2\|\theta_k - \theta'_k\|^2 | \mathcal{F}_k] \leq \mathrm{L}\langle\nabla f(\theta_k) - \nabla f(\tilde{\theta}_k), \theta_k - \tilde{\theta}_k\rangle\|\theta_k - \tilde{\theta}_k\|^2$$

$$\mathsf{E}[\langle\nabla F(\theta_k, \xi_{k+1}) - \nabla F(\tilde{\theta}_k, \xi_{k+1}), \theta_k - \tilde{\theta}_k\rangle\|\theta_k - \tilde{\theta}_k\|^2 | \mathcal{F}_k] = \langle\nabla f(\theta_k) - \nabla f(\tilde{\theta}_k), \theta_k - \tilde{\theta}_k\rangle\|\theta_k - \tilde{\theta}_k\|^2.$$

Similarly,

$$\mathsf{E}[\langle\nabla F(\theta_k, \xi_{k+1}) - \nabla F(\tilde{\theta}_k, \xi_{k+1}), \theta_k - \tilde{\theta}_k\rangle\|\nabla F(\theta_k, \xi_{k+1}) - \nabla F(\tilde{\theta}_k, \xi_{k+1})\|^2 | \mathcal{F}_k]$$
$$\leq \mathrm{L}^2\langle\nabla f(\theta_k) - \nabla f(\tilde{\theta}_k), \theta_k - \tilde{\theta}_k\rangle\|\theta_k - \tilde{\theta}_k\|^2$$

Combining all inequalities above, we obtain

$$\mathsf{E}[\|\theta_{k+1} - \theta'_{k+1}\|^4 | \mathcal{F}_k] \leq \|\theta_k - \tilde{\theta}_k\|^4$$
$$- (4\gamma - \gamma^4\,\mathrm{L}^3 - 4\gamma^2\,\mathrm{L} - 2\gamma^2\,\mathrm{L} - 4\gamma^3\,\mathrm{L}^2)\langle\nabla f(\theta_k) - \nabla f(\tilde{\theta}_k), \theta_k - \tilde{\theta}_k\rangle\|\theta_k - \tilde{\theta}_k\|^2$$

Using A1 and since $1 - \gamma^3\,\mathrm{L}^3/4 - 3\gamma\,\mathrm{L}/2 - \gamma^2\,\mathrm{L}^2 \geq 1 - 11\gamma\,\mathrm{L}/4$, we get

$$\mathsf{E}[\|\theta_{k+1} - \tilde{\theta}_{k+1}\|^4 | \mathcal{F}_k] \leq (1 - 4\gamma\mu(1 - 11\gamma\,\mathrm{L}/4))\|\theta_k - \tilde{\theta}_k\|^4$$
$$\leq (1 - 2\gamma\mu(1 - 11\gamma\,\mathrm{L}/4))^2\|\theta_k - \tilde{\theta}_k\|^4.$$

Since $1 - 11\gamma\,\mathrm{L}/4 \geq 1/2$ for $\gamma \leq 2/(11\,\mathrm{L})$, we complete the proof.

For simplicity of proof for second part of lemma we define $\delta_{k+1} = \|\theta_{k+1} - \theta'_{k+1}\|$. Then we have

$$\mathsf{E}[\delta_{k+1}^{2p} | \mathcal{F}_k]$$
$$= \mathsf{E}[(\delta_k^2 - 2\gamma\langle\nabla F(\theta_k, \xi_{k+1}) - \nabla F(\tilde{\theta}_k, \xi_{k+1}), \theta_k - \tilde{\theta}_k\rangle + \gamma^2\|\nabla F(\theta_k, \xi_{k+1}) - \nabla F(\tilde{\theta}_k, \xi_{k+1})\|^2)^p | \mathcal{F}_k]$$
$$= \mathsf{E}[\sum_{\substack{i+j+l=p; \\ i,j,l\in\{0,\ldots p\}}} \frac{p!}{i!j!l!}\delta_k^{2i}(-2\gamma\langle\nabla F(\theta_k, \xi_{k+1}) - \nabla F(\tilde{\theta}_k, \xi_{k+1}), \theta_k - \tilde{\theta}_k\rangle)^j\gamma^{2l}\|\nabla F(\theta_k, \xi_{k+1}) - \nabla F(\tilde{\theta}_k, \xi_{k+1})\|^{2l} | \mathcal{F}_k].$$

Now we bound each term in the sum above.

1. First, for $i = p, j = 0, l = 0$ the corresponding term in the sum is equal to $\delta_k^{2p}$.

2. Second, for $i = p - 1, j = 1, l = 0$, we have

$$\mathsf{E}[(-2\gamma\langle\nabla F(\theta_k, \xi_{k+1}) - \nabla F(\tilde{\theta}_k, \xi_{k+1}), \theta_k - \tilde{\theta}_k\rangle)|\mathcal{F}_k] = -2\gamma\langle\nabla f(\theta_k) - \nabla f(\tilde{\theta}_k), \theta_k - \tilde{\theta}_k\rangle .$$

3. Third, for $l \geq 1$ or $j \geq 2$ we use Cauchy-Schwartz inequality and get

$$(2\gamma\langle\nabla F(\theta_k, \xi_{k+1}) - \nabla F(\tilde{\theta}_k, \xi_{k+1}), \theta_k - \tilde{\theta}_k\rangle)^j \gamma^{2l}\|\nabla F(\theta_k, \xi_{k+1}) - \nabla F(\tilde{\theta}_k, \xi_{k+1})\|^{2l}$$
$$\leq 2^j\gamma^{j+2l}\delta_k^j\|\nabla F(\theta_k, \xi_{k+1}) - \nabla F(\tilde{\theta}_k, \xi_{k+1})\|^{2l+j} .$$

Moreover using **A**3$(2p)$, we get

$$\mathsf{E}[2^j\gamma^{j+2l}\delta_k^{2i+j}\|\nabla F(\theta_k, \xi_{k+1}) - \nabla F(\tilde{\theta}_k, \xi_{k+1})\|^{2l+j}|\mathcal{F}_k]$$
$$\leq 2^j\gamma^{j+2l}\delta_k^{2p-2}L^{2l+j-1}\langle\nabla f(\theta_k) - \nabla f(\tilde{\theta}_k), \theta_k - \tilde{\theta}_k\rangle .$$

Combining all inequalities above, we obtain

$$\mathsf{E}[\delta_{k+1}^{2p}|\mathcal{F}_k] \leq \delta_k^{2p} - 2p\gamma\langle\nabla f(\theta_k) - \nabla f(\tilde{\theta}_k), \theta_k - \tilde{\theta}_k\rangle\delta_k^{2p-2}$$
$$+ \Big(\sum_{\substack{i+j+l=p; \\ i,j,l\in\{0,\dots p\} \\ j+2l\geq 2}}\frac{p!}{i!j!l!}2^j\gamma^{j+2l}L^{2l+j-1}\Big)\langle\nabla f(\theta_k) - \nabla f(\tilde{\theta}_k), \theta_k - \tilde{\theta}_k\rangle\delta_k^{2p-2} .$$

Since $\gamma \leq \frac{p}{3^p4L}$, we have

$$\mathsf{E}[\delta_{k+1}^{2p}|\mathcal{F}_k] \leq \delta_k^{2p} - \gamma p\langle\nabla f(\theta_k) - \nabla f(\tilde{\theta}_k), \theta_k - \tilde{\theta}_k\rangle\delta_k^{2p-2} .$$

It remains to apply **A**1 together with an elementary bound $(1 - p\mu\gamma) \leq (1 - \gamma\mu)^p$. $\square$

**Lemma 20.** *Assume **A**1, **A**2, **A**3(4), and **C**1(4). Then for any $\gamma \in (0; 1/(\mathrm{L}\,\mathsf{C}_{\mathrm{step},4})]$, any $n \in \mathbb{N}$ and initial distribution $\nu$ it holds*

$$n^{-1}\mathsf{E}_\nu^{1/2}[\|\sum_{k=n+1}^{2n}\{\psi(\theta_k) - \pi_\gamma(\psi)\}\|^2] \lesssim n^{-1}\mathsf{E}_{\pi_\gamma}^{1/2}[\|\sum_{k=n+1}^{2n}\{\psi(\theta_k) - \pi_\gamma(\psi)\}\|^2]$$
$$+ \frac{\mathrm{L}_3(1 - \gamma\mu)^{(n+1)/2}}{n\gamma\mu}\left(\mathsf{E}_\nu^{1/2}[\|\theta_0 - \theta^\star\|^4] + \frac{\mathrm{D}_{\mathrm{last},4}\gamma\tau_4^2}{\mu}\right) .$$

*Proof.* Using the synchronous coupling construction defined in (47) and the corresponding coupling kernel $\mathrm{K}_\gamma$, we obtain that

$$\mathsf{E}_\nu^{1/2}[\|\sum_{k=n+1}^{2n}\{\psi(\theta_k) - \pi_\gamma(\psi)\}\|^2] = (\mathsf{E}_{\nu,\pi_\gamma}^{\mathrm{K}_\gamma}[\|\sum_{k=n+1}^{2n}\{\psi(\theta_k) - \pi_\gamma(\psi)\}\|^2])^{1/2}$$

$$(90)$$

$$\leq \mathsf{E}_{\pi_\gamma}^{1/2}[\|\sum_{k=n+1}^{2n}\{\psi(\tilde{\theta}_k) - \pi_\gamma(\psi)\}\|^2] + (\mathsf{E}_{\nu,\pi_\gamma}^{\mathrm{K}_\gamma}[\|\sum_{k=n+1}^{2n}\{\psi(\theta_k) - \psi(\tilde{\theta}_k)\}\|^2])^{1/2}$$

Applying Minkowski's inequality to the last term and using Lemma 22, we get

$$(\mathsf{E}_{\nu,\pi_\gamma}^{\mathrm{K}_\gamma}[\|\sum_{k=n+1}^{2n}\{\psi(\theta_k) - \psi(\tilde{\theta}_k)\}\|^2])^{1/2} \leq \sum_{k=n+1}^{2n}(\mathsf{E}_{\nu,\pi_\gamma}^K[\|\{\psi(\theta_k) - \psi(\tilde{\theta}_k)\}\|^2])^{1/2}$$

$$\leq \frac{\mathrm{L}_3}{2}\sum_{k=n+1}^{2n}(\mathsf{E}_{\nu,\pi_\gamma}^{\mathrm{K}_\gamma}[c^2(\theta_k, \tilde{\theta}_k)])^{1/2} .$$

Using Hölder's and Minkowski's inequality and applying Lemma 19 , (69) and (70), we obtain

$$(\mathsf{E}_{\nu,\pi_\gamma}^{\mathrm{K}_\gamma}[c^2(\theta_k, \tilde{\theta}_k)])^{1/2}$$

$$\leq (\mathsf{E}_{\nu,\pi_\gamma}^{\mathrm{K}_\gamma}[\|\theta_k - \tilde{\theta}_k\|^4])^{1/4}\big(\mathsf{E}_{\pi_\gamma}^{1/4}[\|\tilde{\theta}_k - \theta^\star\|^4] + \mathsf{E}_\nu^{1/4}[\|\theta_k - \theta^\star\|^4 + \frac{\gamma^{1/2}\tau_2}{\mu^{1/2}}]\big)$$

$$\leq (1 - \gamma\mu)^{k/2}(\mathsf{E}_{\nu,\pi_\gamma}^{\mathrm{K}_\gamma}[\|\theta_0 - \tilde{\theta}_0\|^4])^{1/4}(\mathsf{E}_\nu^{1/4}[\|\theta_0 - \theta^\star\|^4] + \frac{\mathrm{D}_{\mathrm{last},4}^{1/2}\gamma^{1/2}\tau_4}{\mu^{1/2}} + \frac{\gamma^{1/2}\tau_2}{\mu^{1/2}})$$

$$\lesssim (1 - \gamma\mu)^{k/2}\left(\frac{\mathrm{D}_{\mathrm{last},4}\gamma\tau_4^2}{\mu} + \mathsf{E}_\nu^{1/2}\|\theta_0 - \theta^\star\|^4\right)$$

Combining all inequalities above, we get

$$(\mathsf{E}^{\mathsf{K}_\gamma}_{\nu,\pi_\gamma}[\|\sum_{k=n+1}^{2n}\{\psi(\theta_k)-\psi(\theta'_k)\}\|^2])^{1/2} \lesssim \frac{\mathsf{L}_3(1-\gamma\mu)^{(n+1)/2}}{\gamma\mu}\left(\mathsf{E}^{1/2}_\nu[\|\theta_0-\theta^\star\|^4]+\frac{\mathsf{D}_{\text{last},4}\gamma\tau_4^2}{\mu}\right).$$

Substituting the last inequality into (90) we complete the proof. $\qquad\square$

**Lemma 21.** *Assume **A**1, **A**2, **A**3(6), and **C**1(6). Then for any $\gamma \in (0;1/(\mathsf{L}\,\mathsf{C}_{\text{step},6})]$, $n \in \mathbb{N}$, and initial distribution $\nu$, it holds that*

$$n^{-1}\mathsf{E}^{1/2}_\nu\big[\sum_{k=n+1}^{2n}\|\eta(\theta_k)-\pi_\gamma(\psi)\|^2\big] \le n^{-1}\mathsf{E}^{1/2}_\nu\big[\sum_{k=n+1}^{2n}\|\psi(\theta_k)-\pi_\gamma(\psi)\|^2\big]$$
$$+\frac{\mathsf{L}_4(1-\gamma\mu)^{(n+1)/2}}{n\gamma\mu}\mathsf{E}^{1/2}_\nu[\|\theta_0-\theta^\star\|^6]+\frac{\mathsf{L}_4\,\mathsf{D}_{\text{last},6}^{3/2}\gamma^{3/2}\tau_6^3}{3\mu^{3/2}}. \tag{91}$$

*Proof.* Applying the 4-rd order Taylor expansion with integral remainder, we get that

$$\eta(\theta) = \psi(\theta) + \frac{1}{2}\left(\int_0^1 t^2\nabla^4 f(t\theta^\star+(1-t)\theta)\,dt\right)(\theta-\theta^\star)^{\otimes 3}, \tag{92}$$

and using **A**2, we obtain

$$(1/2)\|\left(\int_0^1 t^2\nabla^4 f(t\theta^\star+(1-t)\theta)\,dt\right)(\theta-\theta^\star)^{\otimes 3}\| \le \mathsf{L}_4\|\theta-\theta^\star\|^3. \tag{93}$$

Therefore, combining (92), **A**2, and applying Minkowski's inequality, we get

$$\mathsf{E}^{1/2}_\nu\big[\sum_{k=n+1}^{2n}\|\eta(\theta_k)-\pi_\gamma(\psi)\|^2\big] \le \mathsf{E}^{1/2}_\nu\big[\sum_{k=n+1}^{2n}\|\psi(\theta_k)-\pi_\gamma(\psi)\|^2\big]$$
$$+\frac{\mathsf{L}_4}{6}\sum_{k=n+1}^{2n}\mathsf{E}^{1/2}_\nu[\|\theta_k-\theta^\star\|^6] \tag{94}$$

Applying **C**1(6) for the last term of (94), we get

$$\mathsf{E}^{1/2}_\nu\big[\sum_{k=n+1}^{2n}\|\eta(\theta_k)-\pi_\gamma(\psi)\|^2\big] \lesssim \mathsf{E}^{1/2}_\nu\big[\sum_{k=n+1}^{2n}\|\psi(\theta_k)-\pi_\gamma(\psi)\|^2\big]+\frac{\mathsf{L}_4\,n\mathsf{D}_{\text{last},6}^{3/2}\gamma^{3/2}\tau_6^3}{\mu^{3/2}}$$
$$+\frac{\mathsf{L}_4(1-\gamma\mu)^{3(n+1)/2}}{1-(1-\gamma\mu)^{3/2}}\mathsf{E}^{1/2}_\nu[\|\theta_0-\theta^\star\|^6]. \tag{95}$$

In remains to notice that $(1-\gamma\mu)^{3/2} \le (1-\gamma\mu)$, and the statement follows. $\qquad\square$

We conclude this section with a technical statement on the properties of the function $\psi$ from (73).

**Lemma 22.** *Let $\psi(\cdot)$ be a function defined in (73). Then for any $\theta, \theta' \in \mathbb{R}^d$, it holds that*

$$\|\psi(\theta)-\psi(\theta')\| \le \frac{1}{2}\mathsf{L}_3\,c(\theta,\theta').$$

*Proof.* For simplicity, let us denote $T = \nabla^3 f(\theta^*)$. Hence,

$$\|\psi(\theta)-\psi(\theta')\| \le \frac{1}{2}\|T(\theta-\theta^\star)^{\otimes 2}-T(\theta'-\theta^\star)^{\otimes 2}\|. \tag{96}$$

Note that

$$\|T\| = \sup_{x\ne 0,y\ne 0,z\ne 0}\frac{\sum\limits_{i,j,k}T_{ijk}x_iy_jz_k}{\|x\|\|y\|\|z\|} \ge \sup_{x\ne 0,y\ne 0}\sup_{z\ne 0}\frac{\sum\limits_{k}z_k\sum\limits_{i,j}T_{ijk}x_iy_j}{\|z\|\|y\|\|x\|} = \sup_{x\ne 0,y\ne 0}\frac{\|t(x,y)\|}{\|y\|\|x\|}, \tag{97}$$

where $t(x, y)_k = \sum\limits_{i,j} T_{ijk} x_i y_j$. Therefore, for any $x, y \in \mathbb{R}^d$, it holds that

$$\|t(x,y)\| \le \|x\| \|y\| \|T\| \tag{98}$$

We denote $v = Tx^{\otimes 2} - Ty^{\otimes 2}$. Then

$$v_k = \sum_{i,j} T_{ijk}(x_i x_j - y_i y_j) = \sum_{i,j} T_{ijk}((x_i - y_i)x_j + (x_i - y_i)y_j) =$$
$$\sum_{i,j} T_{ijk}(x_i - y_i)x_j + \sum_{i,j} T_{ijk}(x_i - y_i)y_j, \tag{99}$$

where the first inequality is true since $T_{ijk} = T_{jik}$ by definition of $T$. Combining (98) and (C) and using triangle inequality, we obtain

$$\|v\| \le \|T\| \|x - y\| (\|x\| + \|y\|) \le \|T\| \|x - y\| (\|x\| + \|y\| + \frac{2\sqrt{2}\tau_2\sqrt{\gamma}}{\sqrt{\mu}}).$$

We complete the proof setting $x = \theta - \theta^\star, y = \theta' - \theta^\star$ $\qquad \square$

## D    PROOF OF THEOREM 9

**Theorem 23** (Version of Theorem 9 with explicit constants). *Let $p \ge 2$ and assume A1, A2, A3(3p), and C1(3p). Then for any $\gamma \in (0, 1/(\mathrm{L}\, \mathsf{C}_{\mathrm{step},3p}) \wedge p/(4 \cdot 3^p \mathrm{L})]$, initial distribution $\nu$, and $n \in \mathbb{N}$, the estimator $\bar{\theta}_n^{(RR)}$ defined in (37) satisfies*

$$\mathsf{E}_\nu^{1/p}[\| \mathrm{H}^\star(\bar{\theta}_n^{(RR)} - \theta^\star)\|^p] \le \frac{c_1 \sqrt{\operatorname{Tr} \Sigma_\varepsilon^\star} p^{1/2}}{n^{1/2}} + \frac{c_2 p\tau_p}{n^{1-1/p}} + \frac{\mathsf{C}_{\mathrm{RR},5}}{n\gamma^{1/2}} + \frac{\mathsf{C}_{\mathrm{RR},6}\gamma^{1/2}}{n^{1/2}} + \mathsf{C}_{\mathrm{RR},7}\gamma^{3/2}$$
$$+ \frac{\mathsf{C}_{\mathrm{RR},8}}{n} + \mathcal{R}_5(n, \gamma, \|\theta_0 - \theta^\star\|),$$

*where we have set*

$$\mathsf{C}_{\mathrm{RR},5} = \frac{c_0 \mathsf{D}_{\mathrm{last},p}^{1/2}\tau_p}{\mu^{1/2}}, \quad \mathsf{C}_{\mathrm{RR},6} = \frac{c_0 \mathrm{L}\, \mathsf{D}_{\mathrm{last},p}^{1/2} p\tau_p}{\mu^{1/2}} + \frac{c_0 \mathrm{L}\, \mathsf{D}_{\mathrm{last},2p} p\tau_{2p}^2}{\mu^{3/2}},$$
$$\mathsf{C}_{\mathrm{RR},7} = c_0 \left(\mathsf{C}_1 + \frac{\mathrm{L}\, \mathsf{D}_{\mathrm{last},3p}^{3/2}\tau_{3p}^3}{\mu^{3/2}}\right), \quad \mathsf{C}_{\mathrm{RR},8} = \frac{c_0 \mathrm{L}\, \mathsf{D}_{\mathrm{last},2p}\tau_{2p}}{\mu^2}, \tag{100}$$

$\mathsf{C}_1$ *is defined in (52), and the remainder term $\mathcal{R}_5(n, \gamma, \|\theta_0 - \theta^\star\|)$ is given by*

$$\mathcal{R}_5(n, \gamma, \|\theta_0 - \theta^\star\|) = \frac{c_0(1 - \gamma\mu)^{(n+1)/2}}{\gamma n}\mathsf{E}_\nu^{1/p}[\|\theta_0 - \theta^\star\|^p] + \frac{c_0 \mathrm{L}\, p(1 - \gamma\mu)^{(n+1)/2}}{\mu^{1/2}\gamma^{1/2}n}\mathsf{E}_\nu^{1/p}[\|\theta_0 - \theta^\star\|^p]$$
$$+ \frac{c_0 \mathrm{L}(1 - \gamma\mu)^{(n+1)/2}}{\gamma\mu n}\left(\mathsf{E}_\nu^{1/p}[\|\theta_0 - \theta^\star\|^{2p}] + \frac{\mathsf{D}_{\mathrm{last},2p}\gamma\tau_{2p}^2}{\mu}\right) + \frac{c_0 \mathrm{L}(1 - \gamma\mu)^{(3/2)n}}{\gamma\mu}\mathsf{E}_\nu^{1/p}[\|\theta_0 - \theta^\star\|^{3p}]$$
$$\tag{101}$$

*Proof.* Using the decomposition (76), we obtain that for any $p \geq 2$, it holds that

$$\mathsf{E}_\nu^{1/p}[\| \mathrm{H}^\star(\bar\theta_n^{(RR)} - \theta^\star)\|^p] \lesssim \underbrace{\frac{1}{n}\mathsf{E}_\nu^{1/p}[\|\sum_{k=n+1}^{2n}\varepsilon_{k+1}(\theta^\star)\|^p]}_{T_1} + \underbrace{\frac{1}{\gamma n}\mathsf{E}_\nu^{1/p}[\|\theta_{n+1}^{(\gamma)} - \theta^\star\|^p] + \frac{1}{\gamma n}\mathsf{E}_\nu^{1/p}[\|\theta_{2n+1}^{(\gamma)} - \theta^\star\|^p]}_{T_2}$$

$$+ \underbrace{\frac{1}{\gamma n}\mathsf{E}_\nu^{1/p}[\|\theta_{n+1}^{(2\gamma)} - \theta^\star\|^p] + \frac{1}{\gamma n}\mathsf{E}_\nu^{1/p}[\|\theta_{2n+1}^{(2\gamma)} - \theta^\star\|^p]}_{T_3}$$

$$+ \underbrace{\frac{1}{n}\mathsf{E}_\nu^{1/p}[\|\sum_{k=n+1}^{2n}\varepsilon_{k+1}(\theta_k^{(\gamma)}) - \varepsilon_{k+1}(\theta^\star)\|^p]}_{T_4}$$

$$+ \underbrace{\frac{1}{n}\mathsf{E}_\nu^{1/p}[\|\sum_{k=n+1}^{2n}\varepsilon_{k+1}(\theta_k^{(2\gamma)}) - \varepsilon_{k+1}(\theta^\star)\|^p]}_{T_5} + \underbrace{\|2\pi_\gamma(\psi) - \pi_{2\gamma}(\psi)\|}_{T_6}$$

$$+ \underbrace{\frac{1}{n}\mathsf{E}_\nu^{1/p}[\|\sum_{k=n+1}^{2n}\psi(\theta_k^{(\gamma)}) - \pi_\gamma(\psi)\|^p] + \frac{1}{n}\mathsf{E}_\nu^{1/p}[\|\sum_{k=n+1}^{2n}\psi(\theta_k^{(2\gamma)}) - \pi_{2\gamma}(\psi)\|^p]}_{T_7}$$

$$+ \underbrace{\frac{1}{n}\sum_{k=n+1}^{2n}\mathsf{E}_\nu^{1/p}[\|G(\theta_k^{(\gamma)})\|^p] + \frac{1}{n}\sum_{k=n+1}^{2n}\mathsf{E}_\nu^{1/p}[\|G(\theta_k^{(2\gamma)})\|^p]}_{T_8}.$$

Now we upper bounds the terms above separately. Applying first the Pinelis version of Rosenthal inequality (Pinelis, 1994, Theorem 4.1) together with **A**3(p), we obtain that

$$T_1 \leq \frac{c_1\sqrt{\mathrm{Tr}\,\Sigma_\varepsilon^\star}p^{1/2}}{n^{1/2}} + \frac{c_2 p\tau_p}{n^{1-1/p}}\,.$$

Applying **C**1(p) (which is implied by **C**1(3p)), we obtain that

$$T_2 + T_3 \lesssim \frac{\mathsf{D}_{\mathrm{last},p}^{1/2}\tau_p}{\mu^{1/2}n\gamma^{1/2}} + \frac{(1-\gamma\mu)^{(n+1)/2}}{\gamma n}\mathsf{E}_\nu^{1/p}[\|\theta_0 - \theta^\star\|^p]\,.$$

Applying Lemma 16 (see the bound (72)), we get that

$$T_4 + T_5 \lesssim \frac{\mathsf{L}\,\mathsf{D}_{\mathrm{last},p}^{1/2}\gamma^{1/2}p\tau_p}{\mu^{1/2}n^{1/2}} + \frac{\mathsf{L}\,p(1-\gamma\mu)^{(n+1)/2}}{\mu^{1/2}\gamma^{1/2}n}\mathsf{E}_\nu^{1/p}[\|\theta_0 - \theta^\star\|^p]\,.$$

Using the bounds (77) and (78), we obtain

$$T_6 \lesssim \mathsf{C}_1\gamma^{3/2}\,.$$

Applying Proposition 8, we get

$$\frac{1}{n}\mathsf{E}_{\pi_\gamma}^{1/p}[\|\sum_{k=n+1}^{2n}\psi(\theta_k^{(\gamma)}) - \pi_\gamma(\psi)\|^p] \lesssim \frac{\mathsf{L}\,\mathsf{D}_{\mathrm{last},2p}p\tau_{2p}^2\gamma^{1/2}}{\mu^{3/2}n^{1/2}} + \frac{\mathsf{L}\,\mathsf{D}_{\mathrm{last},2p}\tau_{2p}}{\mu^2 n}\,.$$

Using this bound and Lemma 24, we obtain that

$$T_7 \lesssim \frac{\mathsf{L}\,\mathsf{D}_{\mathrm{last},2p}p\tau_{2p}^2\gamma^{1/2}}{\mu^{3/2}n^{1/2}} + \frac{\mathsf{L}\,\mathsf{D}_{\mathrm{last},2p}\tau_{2p}}{\mu^2 n} + \frac{\mathsf{L}(1-\gamma\mu)^{(n+1)/2}}{\gamma\mu n}\left(\mathsf{E}_\nu^{1/p}[\|\theta_0 - \theta^\star\|^{2p}] + \frac{\mathsf{D}_{\mathrm{last},2p}\gamma\tau_{2p}^2}{\mu}\right)$$

Finally, applying the definition of $G(\theta)$ in (38) together with **C**1(3p), we obtain that

$$T_8 \lesssim \frac{\mathsf{L}\,\mathsf{D}_{\mathrm{last},3p}^{3/2}\gamma^{3/2}\tau_{3p}^3}{\mu^{3/2}} + \frac{\mathsf{L}}{n}\sum_{k=n+1}^{2n}(1-\gamma\mu)^{(3/2)k}\mathsf{E}_\nu^{1/p}[\|\theta_0 - \theta^\star\|^{3p}]$$

$$\lesssim \frac{\mathsf{L}\,\mathsf{D}_{\mathrm{last},3p}^{3/2}\gamma^{3/2}\tau_{3p}^3}{\mu^{3/2}} + \frac{\mathsf{L}(1-\gamma\mu)^{(3/2)n}}{\gamma\mu}\mathsf{E}_\nu^{1/p}[\|\theta_0 - \theta^\star\|^{3p}]\,.$$

To complete the proof it remains to combine the bounds for $T_1$ to $T_8$. $\qquad\square$

### D.1 Proof of Proposition 8

In the proof below we use the notation

$$\bar{\psi}(\theta) = \psi(\theta) - \pi_\gamma(\psi).$$

We proceed with the blocking technique. Indeed, let us set the parameter

$$m = m(\gamma) = \left\lceil \frac{2\log 4}{\gamma\mu} \right\rceil. \tag{102}$$

Our choice of parameter $m(\gamma)$ is due to Proposition 1. For notation conciseness we write it simply as $m$, dropping its dependence upon $\gamma$. Using Minkowski's inequality, we obtain that

$$\mathsf{E}_{\pi_\gamma}^{1/p}\big[\|\sum_{k=0}^{n-1}\bar{\psi}(\theta_k)\|^p\big] \leq \mathsf{E}_{\pi_\gamma}^{1/p}\big[\|\sum_{k=0}^{\lfloor n/m\rfloor m-1}\bar{\psi}(\theta_k)\|^p\big] + m\mathsf{E}_{\pi_\gamma}^{1/p}\big[\|\bar{\psi}(\theta_0)\|^p\big]. \tag{103}$$

Now we consider the Poisson equation, associated with $\mathrm{Q}_\gamma^m$ and function $\bar{\psi}$, that is,

$$g_m(\theta) - \mathrm{Q}_\gamma^m g_m(\theta) = \bar{\psi}(\theta). \tag{104}$$

The function

$$g_m(\theta) = \sum_{k=0}^{\infty} \mathrm{Q}_\gamma^{km}\bar{\psi}(\theta) \tag{105}$$

is well-defined under the assumptions **A**1, **A**2, **A**3$(2p)$, and **C**1$(2p)$. Moreover, $g_m$ is a solution of the Poisson equation (104). Define $q := \lfloor n/m \rfloor$, then we have

$$\sum_{k=0}^{qm-1}\bar{\psi}(\theta_k) = \sum_{r=0}^{m-1} B_{m,r}, \quad \text{with} \quad B_{m,r} = \sum_{k=0}^{q-1}\big\{g_m(\theta_{km+r}) - \mathrm{Q}_\gamma^m g_m(\theta_{km+r})\big\}. \tag{106}$$

Using Minkowski's inequality, we get from (103), that

$$\mathsf{E}_{\pi_\gamma}^{1/p}\big[\|\sum_{k=0}^{n-1}\bar{\psi}(\theta_k)\|^p\big] \leq m\mathsf{E}_{\pi_\gamma}^{1/p}\big[\|\sum_{k=1}^{q}\big\{g_m(\theta_{km}) - \mathrm{Q}_\gamma^m g_m(\theta_{(k-1)m})\big\}\|^p\big] + 2m\mathsf{E}_{\pi_\gamma}^{1/p}\big[\|\psi(\theta_0\|^p\big] \tag{107}$$

Now we upper bound both terms of (107) separately. Under assumption **A**2, and applying **C**1$(2p)$, we get

$$\mathsf{E}_{\pi_\gamma}^{1/p}\big[\|\psi(\theta_0)\|^p\big] \leq \frac{\mathrm{L}}{2}\mathsf{E}_{\pi_\gamma}^{1/p}\big[\|\theta_0 - \theta^\star\|^{2p}\big] \leq \frac{\mathrm{L}\,\mathrm{D}_{\mathrm{last},2p}\gamma\tau_{2p}^2}{2\mu}. \tag{108}$$

To proceed with the first term, we apply Burkholder's inequality (Osekowski, 2012, Theorem 8.6), and obtain that

$$\mathsf{E}_{\pi_\gamma}^{1/p}\big[\|\sum_{k=1}^{q}\big\{g_m(\theta_{km}) - \mathrm{Q}_\gamma^m g_m(\theta_{(k-1)m})\big\}\|^p\big]$$
$$\leq p\mathsf{E}_{\pi_\gamma}^{1/p}\big[\big(\sum_{k=1}^{q}\|\big\{g_m(\theta_{km}) - \mathrm{Q}_\gamma^m g_m(\theta_{(k-1)m})\big\}\|^2\big)^{p/2}\big]. \tag{109}$$

Applying now Minkowski's inequality again, we get

$$\mathsf{E}_{\pi_\gamma}^{2/p}\big[\big(\sum_{k=1}^{q}\|\big\{g_m(\theta_{km}) - \mathrm{Q}_\gamma^m g_m(\theta_{(k-1)m})\big\}\|^2\big)^{p/2}\big] \leq q\mathsf{E}_{\pi_\gamma}^{2/p}\big[\|\big\{g_m(\theta_{km}) - \mathrm{Q}_\gamma^m g_m(\theta_{(k-1)m})\big\}\|^p\big]$$
$$\lesssim q\left(\mathsf{E}_{\pi_\gamma}^{2/p}[\|g_m(\theta_0)\|^p] + \mathsf{E}_{\pi_\gamma}^{2/p}[\|\mathrm{Q}_\gamma^m g_m(\theta_0)\|^p]\right)$$
$$\lesssim q\mathsf{E}_{\pi_\gamma}^{2/p}[\|g_m(\theta_0)\|^p].$$

It remains to upper bound the moment $\mathsf{E}_{\pi_\gamma}^{2/p}[\|g_m(\theta_0)\|^p]$. In order to do this, we first note that due to the duality theorem (Douc et al., 2018, Theorem 20.1.2.), we get that for any $k \in \mathbb{N}$,

$$
\begin{aligned}
\|\mathrm{Q}_\gamma^{mk}\psi(\theta) - \pi_\gamma(\psi)\| &= \sup_{u \in \mathbb{R}^d : \|u\|=1} |\mathrm{Q}_\gamma^{mk}(u^\top\psi(\theta)) - \pi_\gamma(u^\top\psi)| \\
&\leq \frac{1}{2}\,\mathrm{L}_3\,\mathbf{W}_c(\delta_\theta \mathrm{Q}_\gamma^{km}, \pi_\gamma) \\
&\leq 2\,\mathrm{L}_3(1/2)^k \mathbf{W}_c(\delta_\theta, \pi_\gamma)\,,
\end{aligned}
$$

where the last inequality is due to Proposition 1. Hence, applying the definition of $g_m(\theta)$ in (105), we obtain that

$$
\mathsf{E}_{\pi_\gamma}^{1/p}[\|g_m(\theta_0)\|^p] \leq \sum_{k=0}^\infty \mathsf{E}_{\pi_\gamma}^{1/p}[\|\mathrm{Q}_\gamma^{km}\bar\psi(\theta)\|^p] \leq 2\,\mathrm{L}_3 \sum_{k=0}^\infty (1/2)^k \mathsf{E}_{\pi_\gamma}^{1/p}[\{\mathbf{W}_c(\delta_\theta, \pi_\gamma)\}^p]\,.
$$

To control the latter term, we simply apply the definition of $\mathbf{W}_c(\delta_\theta, \pi_\gamma)$ and a cost function $c(\theta, \theta')$ together with $\mathbf{C}1(2p)$, we get

$$
\begin{aligned}
\mathsf{E}_{\pi_\gamma}^{1/p}[\{\mathbf{W}_c(\delta_\theta, \pi_\gamma)\}^p] &\lesssim \left( \int_{\mathbb{R}^d \times \mathbb{R}^d} \|\theta - \theta'\|^p \left( \|\theta - \theta^\star\| + \|\theta' - \theta^\star\| + \frac{\tau_2\sqrt{\gamma}}{\sqrt{\mu}} \right)^p \pi_\gamma(\mathrm{d}\theta)\pi_\gamma(\mathrm{d}\theta') \right)^{1/p} \\
&\leq \left( \int \|\theta - \theta'\|^{2p}\pi_\gamma(\mathrm{d}\theta)\pi_\gamma(\mathrm{d}\theta') \right)^{1/2p} \left( \int \left( \|\theta - \theta^\star\| + \|\theta' - \theta^\star\| + \frac{\tau_2\sqrt{\gamma}}{\sqrt{\mu}} \right)^{2p} \pi_\gamma(\mathrm{d}\theta)\pi_\gamma(\mathrm{d}\theta') \right)^{1/2p} \\
&\lesssim \frac{\mathsf{D}_{\mathrm{last},2p}\tau_{2p}^2\gamma}{\mu}\,.
\end{aligned}
$$

Combining now the bounds above in (109), we get that

$$
\mathsf{E}_{\pi_\gamma}^{1/p}\Big[\Big\| \sum_{k=1}^q \big\{g_m(\theta_{km}) - \mathrm{Q}_\gamma^m g_m(\theta_{(k-1)m})\big\} \Big\|^p\Big] \lesssim \frac{\mathsf{D}_{\mathrm{last},2p}\,p\,\mathrm{L}_3\,\tau_{2p}^2\gamma\sqrt{q}}{\mu}\,, \tag{110}
$$

and, hence, substituting into (103), we get

$$
\mathsf{E}_{\pi_\gamma}^{1/p}\Big[\Big\| \sum_{k=0}^{n-1} \bar\psi(\theta_k)\Big\|^p\Big] \lesssim \frac{\mathsf{D}_{\mathrm{last},2p}\,p\,\mathrm{L}_3\,\tau_{2p}^2\gamma\sqrt{q}\,m}{\mu} + \frac{\mathrm{L}\,\mathsf{D}_{\mathrm{last},2p}\tau_{2p}^2\gamma m}{2\mu}\,. \tag{111}
$$

Now the statement follows from the definition of $m = m(\gamma)$ in (102) and $q = \lfloor n/m \rfloor \leq n/m$.

## D.2 VERSION OF PROPOSITION 8 FOR ARBITRARY INITIAL DISTRIBUTION $\nu$.

In order to prove Theorem 23, we need a generalization of Proposition 8 for arbitrary initial distribution $\nu$. The following result holds:

**Lemma 24.** *Let $\{\tilde\theta_k^{(\gamma)}\}_{k\in\mathbb{N}}$ and $\{\theta_k^{(\gamma)}\}_{k\in\mathbb{N}}$ be defined by the synchronous coupling construction* (47), *where $\tilde\theta_0^{(\gamma)} \sim \pi_\gamma$ and $\theta_0^{(\gamma)} \sim \nu$. Then, under assumptions of Proposition 8, for any $\gamma \in (0, 1/(L\mathsf{C}_{\mathrm{step},6})], n \in \mathbb{N}$ and initial distribution $\nu$, it holds that*

$$
\begin{aligned}
\mathsf{E}_\nu^{1/p}\Big[\Big\| \sum_{k=n+1}^{2n} \{\psi(\theta_k^{(\gamma)}) - \pi_\gamma(\psi)\}\Big\|^p\Big] &\leq \mathsf{E}_{\pi_\gamma}^{1/p}\Big[\Big\| \sum_{k=n+1}^{2n} \{\psi(\tilde\theta_k^{(\gamma)}) - \pi_\gamma(\psi)\}\Big\|^p\Big] \\
&\quad + \frac{c_0\,\mathrm{L}_3(1-\gamma\mu)^{(n+1)/2}}{\gamma\mu}\left( \mathsf{E}_\nu^{1/p}[\|\theta_0^{(\gamma)} - \theta^\star\|^{2p}] + \frac{\mathsf{D}_{\mathrm{last},2p}\gamma\tau_{2p}^2}{\mu}\right),
\end{aligned}
$$

*where $c_0$ is an absolute constant.*

*Proof.* We consider the synchronous coupling contraction defined in (47) and denote by $\mathrm{K}_\gamma$ the corresponding coupling kernel. Hence, we have

$$
\begin{aligned}
\mathsf{E}_\nu^{1/p}\Big[\Big\| \sum_{k=n+1}^{2n} \{\psi(\theta_k) - \pi_\gamma(\psi)\}\Big\|^p\Big] &= \Big(\mathsf{E}_{\nu,\pi_\gamma}^{\mathrm{K}_\gamma}\Big[\Big\| \sum_{k=n+1}^{2n} \{\psi(\theta_k) - \pi_\gamma(\psi)\}\Big\|^p\Big]\Big)^{1/p} \\
&\leq \mathsf{E}_{\pi_\gamma}^{1/p}\Big[\Big\| \sum_{k=n+1}^{2n} \{\psi(\tilde\theta_k) - \pi_\gamma(\psi)\}\Big\|^p\Big] + \Big(\mathsf{E}_{\nu,\pi_\gamma}^{\mathrm{K}_\gamma}\Big[\Big\| \sum_{k=n+1}^{2n} \{\psi(\theta_k) - \psi(\tilde\theta_k)\}\Big\|^p\Big]\Big)^{1/p}\,.
\end{aligned}
$$

It remains to bound the last term in the inequality above. Applying Minkowski's inequality together with Lemma 22, we get

$$\left(\mathsf{E}^{\mathrm{K}_\gamma}_{\nu,\pi_\gamma}\left[\|\sum_{k=n+1}^{2n}\{\psi(\theta_k)-\psi(\tilde{\theta}_k)\}\|^p\right]\right)^{1/p} \le \frac{\mathrm{L}_3}{2}\sum_{k=n+1}^{2n}\left(\mathsf{E}^{\mathrm{K}_\gamma}_{\nu,\pi_\gamma}[c^p(\theta_k,\tilde{\theta}_k)]\right)^{1/p}.$$

Using Hölder's and Minkowski's inequalities together with **C**1($2p$) and Lemma 19, we obtain that

$$\left(\mathsf{E}^{\mathrm{K}_\gamma}_{\nu,\pi_\gamma}[c^p(\theta_k,\tilde{\theta}_k)]\right)^{1/p}$$

$$\le (\mathsf{E}^{\mathrm{K}_\gamma}_{\nu,\pi_\gamma}[\|\theta_k-\tilde{\theta}_k\|^{2p}])^{1/(2p)}\left(\mathsf{E}^{1/(2p)}_{\pi_\gamma}[\|\tilde{\theta}_k-\theta^\star\|^{2p}] + \mathsf{E}^{1/(2p)}_\nu[\|\theta_k-\theta^\star\|^{2p} + \frac{2^{3/2}\gamma^{1/2}\tau_2}{\mu^{1/2}}]\right)$$

$$\le (1-\gamma\mu)^{k/2}(\mathsf{E}^{\mathrm{K}_\gamma}_{\nu,\pi_\gamma}[\|\theta_0-\tilde{\theta}_0\|^{2p}])^{1/(2p)}\left(\mathsf{E}^{1/(2p)}_\nu[\|\theta_0-\theta^\star\|^{2p}] + \frac{2\mathrm{D}^{1/2}_{\mathrm{last},2p}\gamma^{1/2}\tau_2 p}{\mu^{1/2}} + \frac{2^{3/2}\gamma^{1/2}\tau_2}{\mu^{1/2}}\right)$$

$$\lesssim (1-\gamma\mu)^{k/2}\left(\frac{\mathrm{D}_{\mathrm{last},2p}\gamma\tau_{2p}^2}{\mu} + \mathsf{E}^{1/p}_\nu\|\theta_0-\theta^\star\|^{2p}\right)$$

Combining all inequalities above, we get

$$\left(\mathsf{E}^{\mathrm{K}_\gamma}_{\nu,\pi_\gamma}\left[\|\sum_{k=n+1}^{2n}\{\psi(\theta_k)-\psi(\tilde{\theta}_k)\}\|^p\right]\right)^{1/p} \lesssim \frac{\mathrm{L}_3(1-\gamma\mu)^{(n+1)/2}}{\gamma\mu}\left(\mathsf{E}^{1/p}_\nu[\|\theta_0-\theta^\star\|^{2p}] + \frac{\mathrm{D}_{\mathrm{last},2p}\gamma\tau_{2p}^2}{\mu}\right),$$

and the statement follows. $\square$

## E  EXPERIMENTAL DETAILS

We recall the error representation (39), and obtain with simple algebra:

$$\mathrm{H}^\star(\bar{\theta}_n^{(\gamma)}-\theta^\star) + n^{-1}\sum_{k=n+1}^{2n}\varepsilon_{k+1}(\theta^\star) = \frac{\theta^{(\gamma)}_{n+1}-\theta^\star}{\gamma n} - \frac{\theta^{(\gamma)}_{2n+1}-\theta^\star}{\gamma n}$$

$$-\frac{1}{n}\sum_{k=n+1}^{2n}\{\varepsilon_{k+1}(\theta^{(\gamma)}_k)-\varepsilon_{k+1}(\theta^\star)\} - \frac{1}{n}\sum_{k=n+1}^{2n}\psi(\theta^{(\gamma)}_k) - \frac{1}{n}\sum_{k=n+1}^{2n}G(\theta^{(\gamma)}_k). \quad (112)$$

Under **A**3(6), the statistics $\frac{1}{n}\sum_{k=n+1}^{2n}\varepsilon_{k+1}(\theta^\star)$ is a sum of independent random variables, and

$$n^{-2}\mathsf{E}[\|\sum_{k=n+1}^{2n}\varepsilon_{k+1}(\theta^\star)\|^2] = \frac{\mathrm{Tr}\,\Sigma^\star_\varepsilon}{n}.$$

Hence, in order to trace the rate of the second-order terms in (41), it is enough to find the decay rate of the right-hand side in (112). We select different sample sizes $n = 250 \times 2^k$, where $k = 0,\dots,14$, and run the SGD procedure (2) based on the constant step sizes $\gamma$ and $2\gamma$, selecting $\gamma = 1/\sqrt{n}$. Then we construct the associated estimates $\bar{\theta}_n^{(\gamma)}$ and $\bar{\theta}_n^{(2\gamma)}$. We conduct $M = 320$ independent parallel runs to approximate the expectations. Code to reproduce experiments is provided at https://github.com/svsamsonov/richardson_romberg_example.

