# OpenReview forum: "Nonasymptotic Analysis of Stochastic Gradient Descent with the Richardson–Romberg Extrapolation"
_ICLR.cc/2025/Conference — ICLR 2025 Poster_

### Official Review · Reviewer_s57H · 2024-10-27

**Soundness:** 3
**Presentation:** 3
**Contribution:** 3
**Rating:** 6
**Confidence:** 3

**Summary:**

The paper contains an analysis of one-pass SGD on stochastic objective functions:

$$\min\_{\theta \in \mathbb{R}^d} f(\theta), \quad \nabla f(\theta)=\mathrm{E}\_{\xi \sim \mathbb{P}_{\xi}}[\nabla F(\theta, \xi)],$$

This is run with constant step-size $\gamma \asymp 1/\sqrt{n}$ with $n$ the number of samples,  and it is shown that $p$-th moments satisfy a bound

$$\mathrm{E}^{1 / p}\left[\left\|\bar{\theta}\_n^{(R R)}-\theta^{\star}\right\|^p\right] \leq \frac{C p^{1 / 2} \sqrt{\operatorname{Tr} \Sigma_{\infty}}}{n^{1 / 2}}+\frac{C(f, d, p)}{n^{3 / 4}}+\mathcal{R}\left(\left\|\theta_0-\theta^{\star}\right\|, n, p\right),$$

Here
* $\bar{\theta}\_n^{(R R)}$ is the ``Richardson-Romburg'' estimator for reducing lower order terms (specifically achieving the $n^{3/4}$ correction.
* $\mathcal{R}$ is an error term that controls the dependence on initial condition.

This improves the correction term in existing work from $n^{1/2+\delta}$ to $n^{3/4}$.

**Strengths:**

This improves an existing bound in various ways, both in the sense of
(1) achieving moment bounds and
(2) improving the rate of the correction term from $n^{1/2+\delta}$ to $n^{3/4}$, which is claimed as optimal.

It does it using the method of the `Richardson-Romberg' extrapolation.

For more common Polyak-Ruppert averaging, it also achieves the same rate provided the minimization is strongly convex.

**Weaknesses:**

1) This paper performs a detailed mathematical analysis of a correction term to the behavior of SGD on fixed problems.  It is hard to imagine important practical conclusions of this -- we are talking about improving subleading correction terms in an _asymptotic_ setting when the number of samples overwhelms all the important problem-dependent constants in a stochastic optimization context.  And yes, I use the term _asymptotic_ intentionally: the $n$ in this problem has to overwhelm constants which are _quite clearly_ not effective, and only once that happens will the subleading terms will have improved dependence.   The authors have _not_ attempted to demonstrate that there are settings where this theorem is experimentally realizable, nor honestly, do I think it would be reasonable to expect them to do so (if they can show their 3/4-power in an experiment, I'd retract this and happily ``eat crow'').

2) The 3/4 power seems to be the essential point of this story.  It is asserted at multiple places that the 3/4 is optimal, but there is no clear construction.  There is a citation to another paper (Li et al), which considers a different algorithm.  If the example from that other paper works, it should be either displayed or precisely cited here (it was not possible to find it in a quick perusal).

**Questions:**

1) As stated: can the authors show that they can realize this n^{3/4} correction term in an experiment, (the setup should be such that I can reproduce it and, the 3/4 fit cleanly on a log-log plot)?

2) The optimality of the 3/4 is asserted multiple places, but it is nowhere displayed.  What construction demonstrates the non-improvability of the 3/4?

---

> ### Author Response · Authors · 2024-11-28
> **Response to reviewer s57H**
>
> ***As stated: can the authors show that they can realize this n^{3/4} correction term in an experiment, (the setup should be such that I can reproduce it and, the 3/4 fit cleanly on a log-log plot)?***
>
> Thank you for your question. We added to the revised version of the paper the simple $1-$dimensional example, which demonstrates the scaling of the second-order term $n^{-3/4}$.  We also provide the code at the anonymized github (see the revised version of our paper) in order to simplify reproducibility of our results. We also attach reference here:
>
> https://anonymous.4open.science/r/richardson_romberg_example-3DD4/
>
> ***The optimality of the 3/4 is asserted multiple places, but it is nowhere displayed. What construction demonstrates the non-improvability of the 3/4?***
>
> In our work, for the Polyak-Ruppert averaging algorithm and Richardson-Romberg extrapolation, we establish an estimate of the form
> $$\mathbb{E}{\nu}[|H (\bar{\theta}{n}^{(RR)} - \theta^\star)|^2] \leq \frac{\operatorname{tr}(\Sigma_\varepsilon)}{n} + O\left(\frac{1}{n^{3/2}}\right),$$
> which demonstrates a clear separation of scales between the leading term and the remainder terms.
> In the paper ROOT-SGD: Sharp Nonasymptotics and Asymptotic Efficiency in a Single Algorithm by Chris Junchi Li, Wenlong Mou, Martin Wainwright, and Michael Jordan (Proceedings of the Thirty-Fifth Conference on Learning Theory, PMLR 178:909–981, 2022), it is stated on page 3, after formula (3), that the second term in ROOT SGD algorithm is unavoidable under a natural setup. Our numerical experiment also shows that the second order term in our algorithm behaves as $n^{-3/2}$ as well. Following your suggestion, we have removed the optimality statement, replacing it with the best known order at the moment. Plese, see revised pdf.

---

> > ### Comment · Reviewer_s57H · 2024-11-29
> >
> > The authors have demonstrated their results in a simple setup.
> >
> > On considering that the order of the 2nd order term can be sharp, and that the paper reproduces an the estimate of Li et al with a simpler algorithm, I have increased my score to a 6.

---

### Official Review · Reviewer_LgiJ · 2024-11-03

**Soundness:** 4
**Presentation:** 3
**Contribution:** 2
**Rating:** 6
**Confidence:** 4

**Summary:**

The authors provide convergence analysis of both the Polyak-Ruppert averaged SGD (PR-SGD) and the Richardson-Romberg extrapolated SGD (RR-SGD) by a two-term expansion of the quantity $ \mathbb E^{1/2}[ || \hat \theta - \theta^* ||^2  ]$, a quantitative refinement of asymptotic normality. They show that RR-SGD achieves the same rate as PR-SGD which is known to be optimal.

**Strengths:**

- The paper is well-motivated and well-written. In particular, the last-iterate bound for RR-SGD, despite some issues raised below, is novel.

**Weaknesses:**

See Questions

**Questions:**

1. The extensive use of $\lesssim$ is rather confusing and potentially undermining in the paper. For example, in Corollaries 4 and 7, are there multiplicative constants suppressed on the leading $n^{-1/2} \sqrt{ \operatorname{tr} \Sigma }$ terms?

* If there are: (1) these corollaries do not match in sharpness the existing results, and provide little insight into how RR-SGD compares to PR-SGD as the latter is favorable exactly because of its Cramer-Rao-type optimality in the leading term.
(2) is it possible to spell out and compare the explicit leading terms?

* If there are not, I wonder why $ \lesssim$ instead of $\le$ is used.

Given how the notation $\lesssim$ is defined in Line 118 I'm afraid the former is the case, right?

Either way, the authors might want to say a few words on what terms/constants are being suppressed at least for presentation purposes.


----


2. Along the same line, we know that (considering decreasing step sizes for the moment --- you might be able to work back and forth between these two regimes) with PR-SGD the asymptotic normal distribution one can prove for
$\sqrt{t} (\widehat{\theta}_{t}^{\mathsf{avg}} - \theta^*)$
has a variance that matches the Cramer-Rao lower bound,
which happens for any step sizes $ \gamma_t \propto t^{-w}$ where $w\in[1/2, 1]$.  Whereas for SGD without averaging, the asymptotic normality takes the form of

$t^{w - 1/2} ( \widehat{\theta}_{t} - \theta^* ) $ where $w$ is the decay rate of step sizes defined above. Even in the $ w= 1 $ case, one gets a variance that's strictly larger than Cramer-Rao (hence than that of the iterate average). When $ w<1 $ this is a strictly slower rate of convergence.

Long story short, PR averaging accelerates the optimization via a "removal of the step size effect". Is it the case that RR has a similar effect without averaging?

---

> ### Author Response · Authors · 2024-11-28
> **Response to reviewer LgiJ**
>
> We thank the reviewer for their valuable feedback and are happy to provide further details in response to the questions they raised.
>
> First, the referee is indeed correct, that the $\lesssim$ symbol is overused in the current submission. However, when considering the leading term of the bounds for the second moment (e.g. in Corollaries 4 and 7), the constant in front of the leading term in our variance bounds is precisely $1$. In case of the $p$-th moment bound the leading term of our bounds is affected by a numerical absolute constant, which comes through the Pinelis version of Rosenthal inequality, see [2], Theorem 4.1. We have revised presentation of our main results in order to reflect this fact.
>
> Second, we would like to bring the referee's attention to the point that our Richardson-Romberg corrected estimator is the corrected version of the Polyak-Ruppert averaged estimator. It is a bit tricky to understand, if RR can help in improving the performance of solely the last iterate of SGD. Indeed, the standard analysis of the last iterate $\theta_n$ in case of constant step size SGD reveals (see e.g. [1]), that
> $$\mathbb{E} [\|\|\theta_n - \theta^{\star}\|\|^2] \leq C_0 \gamma + C_1 \exp^{-\gamma n} \|\theta_0 -  \theta^{\star}\|\|^2\.$$
> Thus, when the number of iterations $n$ is large enough, the fluctuations of $\theta_n$ around $\theta^*$ scales as $\sqrt{\gamma}$, while its bias scales as $\mathcal{O}(\gamma)$. Thus it is hard to expect that applying the RR procedure solely to the last iterate can improve the performance of the algorithm.
>
> ***References***
>
> [1] Aymeric Dieuleveut, Alain Durmus, and Francis Bach. Bridging the gap between constant step size
> stochastic gradient descent and Markov chains. The Annals of Statistics, 48(3):1348 – 1382, 2020.
> doi: 10.1214/19-AOS1850. URL https://doi.org/10.1214/19-AOS1850.
>
> [2] Iosif Pinelis. Optimum Bounds for the Distributions of Martingales in Banach Spaces. The Annals
> of Probability, 22(4):1679 – 1706, 1994. doi: 10.1214/aop/1176988477. URL https://doi.
> org/10.1214/aop/1176988477

---

> > ### Comment · Reviewer_LgiJ · 2024-11-28
> >
> > Thank you for the detailed response. My initial score was already based on my understanding that the $\lesssim$ symbols did not suppress the leading constants. So I would like to keep my scores unchanged and wish the authors the best of luck.
> >
> >
> > Regards,
> >
> > LgiJ

---

### Official Review · Reviewer_UgHP · 2024-11-03

**Soundness:** 3
**Presentation:** 2
**Contribution:** 3
**Rating:** 6
**Confidence:** 3

**Summary:**

The paper considers the minimization of an objective function through unbiased estimates of its gradients. It brings tools from the theory of Markov chains to the study of stochastic gradient descent, showing that, under suitable assumptions, the Richardson-Romberg extrapolation achieves the optimal non-asymptotic risk bound (proved by Li et al. 2022 for the Root-SGD algorithm). Extensions to the p-th moment bound are also provided.

**Strengths:**

The paper interprets SGD updates with constant step sizes as an homogenous Markov chain, and uses a trick from numerical analysis to show that the difference of the average processes obtained for step sizes $\gamma$ and $2\gamma$ (defined in eq. (37)) will converge to a point that is closer to the optimum (as shown by Dieuleveut et al. 2018).

**Weaknesses:**

The paper could use some more polishing in its presentation:
* The geometric ergodicity of the sequence of updates is a crucial aspect, yet is not defined in the preliminaries. It would also be valuable to add an intuitive explanations for how this fact would be used to derive the risk bounds.
* $D_{{\rm last}, 2}$ used eq. (19) in line 222 is not defined.
* In line 480, the beginning of the sentence should be "Directions ..."
* The authors should also make explicit the fact that the number of samples, $n$, needs to be known a priori to optimize the step size $\gamma$.

**Questions:**

Could the authors provide a geometric interpretation for the definition of $c(\theta, \theta')$, and the ensuing geometric ergodicity of the SGD iterates with respect to $W_c$? Also, are the authors aware of other contexts were such non-standard distance-like functions have been used, which could be cited in the paper?

---

> ### Author Response · Authors · 2024-11-28
> **Response to reviewer UgHP**
>
> We thank the reviewer for their valuable feedback and are happy to provide further details in response to the questions they raised.
>
> ***The geometric ergodicity of the sequence of updates is a crucial aspect, yet is not defined in the preliminaries. It would also be valuable to add an intuitive explanations for how this fact would be used to derive the risk bounds.***
>
> We thank the reviewer for their suggestion. They are, of course, correct in emphasizing that the geometric ergodicity of the update is crucial to our derivation. In fact, the geometric ergodicity of the iterates $\\{\theta_{k}^{(\gamma)}\\}$ is stated in the Proposition 1. For convenience, we have revised its statement accordingly. The geometric ergodicity of this chain (or, equivalently, its underlying Markov kernel) is used in the proof of the Rosenthal inequality, which we state in Proposition 8 of our submission. This inequality enables us to obtain sharp bounds on the terms in the decompositions outlined in Equations (31) and (39). The intuition here is the fact that the covariances between the elements $\theta_{k}^{(\gamma)}$ and $\theta_{k+\ell}^{(\gamma)}$ decays exponentially with the growth of $\ell$, given that $\\{\theta_{k}^{(\gamma)}\\}$ is geometrically ergodic. We included this explanation in the revised version of our paper.
>
> *** In line 480, the beginning of the sentence should be "Directions …” $D_{\last,2}$ used eq. (19) in line 222 is not defined.***
>
> We thank the reviewer for pointing out these two omissions, which we will address in the revision of our manuscript.
>
> ***The authors should also make explicit the fact that the number of samples, $n$, needs to be known a priori to optimize the step size $\gamma$.***
>
> We will add this remark to the revised version of our manuscript, starting from the introduction where we outline our contributions.
>
> ***Could the authors provide a geometric interpretation for the definition of $c(\theta,\theta’)$, and the ensuing geometric ergodicity of the SGD iterates with respect to $W_c$? Also, are the authors aware of other contexts were such non-standard distance-like functions have been used, which could be cited in the paper?***
>
> We can distinguish two behaviors in the distance-like function $ c $ depending on $ \Vert \theta - \theta' \Vert $. If $ \Vert \theta - \theta^\star \Vert \vee \Vert \theta - \theta^\star \Vert \geq \frac{2\sqrt{2} \tau_2 \sqrt{\gamma}}{\mu} $, $ c(\theta, \theta') $ behaves essentially as $ \Vert \theta - \theta' \Vert \times(  \Vert \theta - \theta^\star \Vert \vee \Vert \theta - \theta^\star \Vert) $. However, if $ \Vert \theta - \theta' \Vert \leq \frac{2\sqrt{2} \tau_2 \sqrt{\gamma}}{\mu} $, the dominating term in the definition of $ c $ is $ \frac{2\sqrt{2} \tau_2 \sqrt{\gamma} \Vert \theta - \theta' \Vert}{\mu} $. Note that convergence in $ W_2 $ has been shown in [1], and initially, we considered relying on convergence in $ W_2 $ since $ c $ is roughly equivalent to the squared norm. However, this led to suboptimal bounds with respect to the step size, which is why we designed the distance-like function $ c $.
> Indeed, the main justification for introducing this distance is that we need to bound terms of the form $ \vert Q_{\gamma}^k f(\theta) - \pi_{\gamma}(f) \vert $, for some functions $ f $ satisfying $ \vert f(\theta) - f(\theta') \vert \leq c(\theta, \theta') $, but not $ \vert f(\theta) - f(\theta') \vert \leq \Vert \theta - \theta' \Vert $, such as the function $ \psi $ defined in our paper. Therefore, convergence in $ W_2 $ is not sufficient or leads to suboptimal bounds.

---

### Official Review · Reviewer_8ENV · 2024-11-03

**Soundness:** 3
**Presentation:** 3
**Contribution:** 3
**Rating:** 6
**Confidence:** 2

**Summary:**

The paper investigates the convergence of stochastic gradient descent (SGD) for strongly convex and smooth problems. Concretely, using the mean squared distance as the performance metric, the author proves that the Polyak-Ruppert averaging procedure with the Richardson-Romberg extrapolation technique can achieve the optimal convergence rate for both the leading and higher-order terms.

**Strengths:**

1. The authors provide interesting results on SGD with Richardson-Romberg extrapolation.
2. I also appreciate the authors' honest discussion of their stronger assumption compared to the previous works, e.g., the higher-order smoothness.
3. Moreover, the writing is easy to follow.

**Weaknesses:**

Since I am not very familiar with related works of the Richardson-Romberg extrapolation, the author could feel free to point out anything wrong in my following question.

1. Could the authors provide more discussion on C1? Especially, when can (20) hold?

2. The paper studies the convergence in expectation, is it possible to establish the high-probability convergence result with the same rate for both leading and higher-order terms?

3. One point I found is that when the stochastic problem degenerates into the deterministic problem, the current rate cannot recover the well-known linear convergence result. Could the authors make some statement on this point?

4. Another thing I am interested in is that the leading term $O(n^{-1/2})$ seems not to be infected by the stepsize (even if without the Richardson-Romberg extrapolation), which seems very different from the existing analysis of SGD for the same problem. Could the authors elaborate more on this fact?

**Questions:**

See **Weaknesses** part.

---

> ### Author Response · Authors · 2024-11-28
> **Response to reviewer 8ENV**
>
> We thank the reviewer for their feedback and we are happy to provide further details on the questions they formulated.
>
> ***Could the authors provide more discussion on C1? Especially, when can (20) hold?***
>
> It is shown in [1, Lemma 13] that $ C_1(p) $ (and not only (21)) holds under the assumptions A1, A2, and A3(p) in our papers. The main reason we set $ C_1(p) $ as an assumption in our work is that, while this result is valid, we believe the constants appearing in [1, Lemma 13] are likely suboptimal. Specifically, $ D_{last,p} = O(p!!) $ when using [1, Lemma 13]. While we consider these constants suboptimal, deriving tighter bounds is challenging and we believe it falls outside the scope of our current submission. We leave this for future work, along with deriving tight high-probability bounds for RR extrapolation (as mentioned in the answer to your second question).
>
> ***The paper studies the convergence in expectation, is it possible to establish the high-probability convergence result with the same rate for both leading and higher-order terms?***
>
> Based on our $ L^p $ bounds (Theorem 9 in our main paper), we can establish high-probability bounds using Markov's inequality for some adjusted moment $ p $. Typically, this moment is tuned with respect to the desired excess probability in order to obtain the tightest bounds. However, since the constants in our results depend on those in $ C_1(p) $, our bounds remain implicit. Furthermore, directly incorporating the bounds from [1, Lemma 13] would lead to overly pessimistic estimates. As previously emphasized, the derivation of tight constants in $ C_1(p) $ and tight high-probability bounds are closely related, and for this reason, we have decided to leave these two problems for future work.
>
> ***One point I found is that when the stochastic problem degenerates into the deterministic problem, the current rate cannot recover the well-known linear convergence result. Could the authors make some statement on this point?***
>
> We can actually recover the linear convergence result if we consider the deterministic problem. In this case, the noise $ \varepsilon $ is simply zero, and $ C_1(p) $ is satisfied for any $ p $, with $ D_{last,p} = 0 $. Therefore, Theorems 3 and 6 provide exponential convergence bounds, which are embedded in the remainder term. Indeed, all other terms are zero: the covariance matrix $ \Sigma_{\varepsilon} $ is zero because $ \varepsilon $ is zero, and the other terms are proportional to some $ D_{last,p} $ for some $ p $, and are therefore also zero. We will include this remark in the revised version of the paper and thank the reviewer for their insightful question.
>
> ***Another thing I am interested in is that the leading term $O(n^{-1/2})$  seems not to be infected by the stepsize (even if without the Richardson-Romberg extrapolation), which seems very different from the existing analysis of SGD for the same problem. Could the authors elaborate more on this fact?***
>
> Indeed, some works have derived a leading term of $ O(n^{-1/2}) $, which depends on the step size. For example, in [1, Corollary 6], which we extend and improve, the authors also provide an expansion for the number of iterations with a fixed step size, applied to the expectation of $ (\bar{\theta}_k^{(\gamma)} - \theta^\star) $. However, when closely examining their result, one can see that the leading term takes the form
>
> $$ n^{-1/2} \int \mathcal{C}(\tilde{\theta}) \, \mathrm{d} \pi_{\gamma}(\tilde{\theta}), $$
>
> where $ \mathcal{C}(\tilde{\theta}) $ is the covariance matrix of $ \varepsilon_1(\tilde{\theta}) $. In fact, we show in our paper that
>
> $$ \int \mathcal{C}(\tilde{\theta}) \, \mathrm{d} \pi_{\gamma}(\tilde{\theta}) = \mathcal{C}(\tilde{\theta}) + O(\gamma), $$
>
> which leads to a leading term that is not affected by the step size.
>
> ***References***
>
> [1] Aymeric Dieuleveut, Alain Durmus, and Francis Bach. Bridging the gap between constant step size
> stochastic gradient descent and Markov chains. The Annals of Statistics, 48(3):1348 – 1382, 2020.
> doi: 10.1214/19-AOS1850. URL https://doi.org/10.1214/19-AOS1850.

---

> > ### Comment · Reviewer_8ENV · 2024-11-29
> >
> > I thank the authors' detailed response. I would like to maintain my score since, as stated, I am not familiar with the Richardson-Romberg extrapolation. However, I will increase my contribution score to 3.

---

### Author Response · Authors · 2024-11-28
**Paper Revision**

We highly appreciate the valuable feedback provided by all reviewers. In the revised version, we have made every effort to address all the remarks. In particular, the Reviewer s57H emphasized the need to justify the optimality of the second-order term.
In our work, we proposed an algorithm based on the use of Polyak-Ruppert averaging and Richardson-Romberg extrapolation. For this algorithm, we derived an optimal estimate for the MSE and obtained a second-order term that is currently the best known and matches the second-order term obtained in prior work ROOT-SGD: Sharp Nonasymptotics and Asymptotic Efficiency in a Single Algorithm by Chris Junchi Li, Wenlong Mou, Martin Wainwright, Michael Jordan Proceedings of Thirty Fifth Conference on Learning Theory, PMLR 178:909-981, 2022. It is worth noting that our algorithm is significantly simpler and does not require the construction of control variates. A carefully designed numerical experiment demonstrates that the order of the second term cannot be improved in the general case, which suggests its optimality. We leave the derivation of a lower bound for future work.

Main changes in the new pdf are highlighted in blue.

---

### Meta-Review · Area_Chair_atco · 2024-12-21

**Metareview:**

This paper considers constant step size SGD on strongly convex and smooth objectives and provides a Markov chain based analysis. Authors derive a bias-variance decomposition of the MSE with a dominant term and a higher term of respective orders 1/n^.5 and 1/n^.75.


This paper was reviewed by four expert reviewers and received the following Scores/Confidence: 6/3, 6/3, 6/4, 6/2. I think the paper is studying an interesting topic and the results are relevant to ICLR community. The following concerns were brought up by the reviewers:

- There are some important and relevant references on the analysis of "constant step size SGD" missing. Authors should include references to existing works on this algorithm.

- The optimality of power 3/4 is not explained clearly in the paper. Examples would help.

- Several typos and ambiguous statements/notations were pointed by the reviewers. These should be carefully addressed.

- Certain technical terminology are not clearly defined in the paper. While I admit not all terms can be included in the paper, I suggest authors to make their paper more friendly to non-experts on MCMC.


Authors should carefully go over reviewers' suggestions and address any remaining concerns in their final revision.  Based on the reviewers' suggestion, as well as my own assessment of the paper, I recommend including this paper to the ICLR 2025 program.

**Additional Comments On Reviewer Discussion:**

Reviewer questions are thoroughly answered by the authors. The revision provides color-coded updates.

---

### Decision · Program_Chairs · 2025-01-22

Accept (Poster)